# Dopamine and serotonin in human substantia nigra track social context and value signals during economic exchange

**Seth R. Batten** [1,15] ✉, **Dan Bang** [1,2,3,4,15] ✉, **Brian H. Kopell**[5,6,15], **Arianna N. Davis** [7,8], **Matthew Heflin** [7,8], **Qixiu Fu**[7,8], **Ofer Perl**[7,8], **Kimia Ziafat**[7], **Alice Hashemi** [7], **Ignacio Saez** [7], **Leonardo S. Barbosa**[1], **Thomas Twomey**[1], **Terry Lohrenz** [1], **Jason P. White** [1], **Peter Dayan** [9,10], **Alexander W. Charney**[7], **Martijn Figee**[5,6,7], **Helen S. Mayberg** [5,6], **Kenneth T. Kishida** [11,12], **Xiaosi Gu** [7,8,13] ✉ **& P. Read Montague** [1,3,14] ✉

Dopamine and serotonin are hypothesized to guide social behaviours. In humans, however, we have not yet been able to study neuromodulator dynamics as social interaction unfolds. Here, we obtained subsecond estimates of dopamine and serotonin from human substantia nigra pars reticulata during the ultimatum game. Participants, who were patients with Parkinson's disease undergoing awake brain surgery, had to accept or reject monetary offers of varying fairness from human and computer players. They rejected more offers in the human than the computer condition, an effect of social context associated with higher overall levels of dopamine but not serotonin. Regardless of the social context, relative changes in dopamine tracked trial-by-trial changes in offer value—akin to reward prediction errors—whereas serotonin tracked the current offer value. These results show that dopamine and serotonin fluctuations in one of the basal ganglia's main output structures reflect distinct social context and value signals.

Successful social interaction requires us to infer other people's mental states, such as their desires and intentions, and adapt our behaviour to meet social norms for what is appropriate and fair[1–3]. Our understanding of how the human brain supports social interaction is mainly based on non-invasive methods for functional neuroimaging, such as electro-encephalography and functional magnetic resonance imaging (fMRI).

While they have helped map a network of brain regions, the 'social brain', which is activated by social tasks[4,5], these methods typically involve a trade-off between spatial and temporal resolution and provide data that reflect a mixture of electrical and chemical events[6,7]. As a result, we know little about the human social brain at finer biological scales. One recent study in brain surgery patients reported single-neuronal

[1]Fralin Biomedical Research Institute at VTC, Virginia Tech, Roanoke, VA, USA. [2]Center of Functionally Integrative Neuroscience, Aarhus University, Aarhus, Denmark. [3]Wellcome Centre for Human Neuroimaging, University College London, London, UK. [4]Department of Experimental Psychology, University of Oxford, Oxford, UK. [5]Nash Family Center for Advanced Circuit Therapeutics, Icahn School of Medicine at Mount Sinai, New York, NY, USA. [6]Center for Neuromodulation, Department of Neurosurgery, Icahn School of Medicine at Mount Sinai, New York, NY, USA. [7]Department of Psychiatry, Icahn School of Medicine at Mount Sinai, New York, NY, USA. [8]Center for Computational Psychiatry, Icahn School of Medicine at Mount Sinai, New York, NY, USA. [9]Max Planck Institute for Biological Cybernetics, Tübingen, Germany. [10]University of Tübingen, Tübingen, Germany. [11]Department of Translational Neuroscience, Wake Forest School of Medicine, Winston-Salem, NC, USA. [12]Department of Neurosurgery, Wake Forest School of Medicine, Winston-Salem, NC, USA. [13]Nash Family Department of Neuroscience, Icahn School of Medicine at Mount Sinai, New York, NY, USA. [14]Department of Physics, Virginia Tech, Blacksburg, VA, USA. [15]These authors contributed equally: Seth R. Batten, Dan Bang, Brian H. Kopell. ✉e-mail: srbatten10@vtc.vt.edu; danbang@cfin.au.dk; xiaosi.gu@mssm.edu; read@vtc.vt.edu

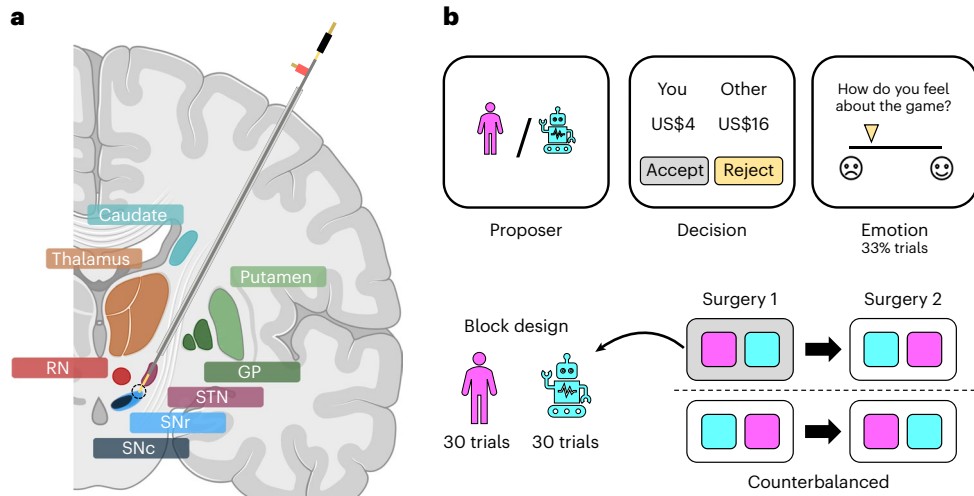

**Fig. 1 | Experimental framework. a**, Illustration of electrode trajectory and recording site. GP, globus pallidus; STN, subthalamic nucleus; SNr/SNc, substantia nigra pars reticulata/compacta; RN, raphe nucleus. Brain slice created with Biorender.com. **b**, The game involved 60 trials of the one-shot ultimatum game where participants had to accept or reject splits of a US$20 stake proposed by either a human (30 trials) or a computer (30 trials) avatar. On around a third of trials, participants were asked to indicate how they felt about the game. Each participant underwent two surgical sessions 14–28 days apart; the human and computer conditions were blocked within a session and their order was counterbalanced across sessions.

electrophysiological correlates of social reasoning[8], but a similar breakthrough has yet to be made for the neuromodulatory systems that regulate activity across the social brain.

Our knowledge about the roles of neuromodulators such as dopamine and serotonin in human social interaction is mainly derived from combining pharmacology with economic games[9]. One set of studies manipulated serotonin levels during the ultimatum game[10,11], a two-person 'take-it-or-leave-it' game probing social fairness norms[12]. A 'proposer' offers a split of a monetary stake (for example, US$20) to a 'responder' who can then accept or reject the split. The proposer can make any offer, from keeping to sharing the full stake. If the responder accepts the offer, both parties get the suggested amounts. If the responder rejects the offer, both parties get nothing. In Western cultures, responders are more likely to reject offers equal to or lower than 20% of the total stake (for example, US$4 out of US$20)[13]. Intriguingly, when their serotonin levels are lowered by dietary acute tryptophan depletion, people reject almost all 'unfair' offers[10]. Conversely, when their serotonin levels are heightened by selective serotonin reuptake inhibitors, people reject fewer unfair offers[11]. A related study found that levodopa, which increases dopamine levels, makes people less averse to inflicting pain on others in exchange for money[14].

These studies have established causal roles for neuromodulators in human social interaction, but pharmacology, which has limited signal resolution, cannot resolve their contribution at fast timescales. Using invasive methods such as single-unit recordings and optogenetic activation, animal studies have shown that fast dopamine and serotonin fluctuations carry signals that are critical for adaptive behaviour[15–19]. For example, transient changes in dopamine are believed to reflect reward prediction errors (RPEs)—the difference between actual and expected reward—which can be used to learn the value of stimuli or actions for future behavioural control[15–17]. This computational motif for dopamine has recently been extended to learning the value of conspecifics in mice[20]. Similarly, transient changes in serotonin have been implicated in the coding of both non-social and social rewards[18,19,21]. Yet animal models of social interaction cannot fully represent the complexity of their human counterparts, and it remains unknown how fast neuromodulator signalling contributes to uniquely human aspects of social interaction.

We have recently developed an approach for measuring fast neuromodulator fluctuations in the conscious human brain[22–26]. This opportunity arises in patients undergoing awake brain surgery for the implantation of a deep brain stimulation (DBS) electrode for the management of disease symptoms (for example, movement disorder symptoms in Parkinson's disease or essential tremor). With minimal changes from standard operating procedures, a carbon-fibre electrode can be inserted into deep structures of the brain and used to make electrochemical recordings during experimental tasks. To date, this approach, human electrochemistry, has shown that dopamine and serotonin in the human striatum track RPEs in a reward-based task[23,24,26] and carry both sensory and action-related information during perceptual decision-making[25]. Yet it remains unknown whether these computational roles are modulated by social context.

Here, we address these questions, by using electrochemistry to measure fast dopamine and serotonin fluctuations during a social task. Specifically, participants were Parkinson's disease patients undergoing DBS surgery and played the role of the responder in the ultimatum game. The recording site is naturally constrained by the surgical procedure; here, it was substantia nigra pars reticulata (SNr). The SNr is one of the basal ganglia's main output nuclei[27]; it receives projections from dopamine neurons in the substantia nigra pars compacta (SNc)[28] and serotonin neurons in the dorsal raphe nucleus[29]. To directly probe the effects of social context, participants were told that the proposers were other people in one half of the trials and a computer in the other half of the trials. In line with previous results[30–33], participants rejected more human than computer offers. Mirroring this effect of social context, overall levels of dopamine, but not serotonin, were higher in the human condition. Changes in dopamine relative to a local baseline tracked trial-by-trial changes in offer value—akin to RPE signalling—whereas relative changes in serotonin tracked the current offer value irrespective of the social context.

## Results

### Experimental framework

Parkinson's disease patients ($n = 4$) undergoing DBS surgery performed the ultimatum game while we obtained electrochemical estimates of dopamine and serotonin in the SNr (Fig. 1a). Our electrochemistry protocol, which builds on earlier work in both animals[34,35] and humans[22–26], provides ten samples per second. In brief, the protocol involves the repeated delivery of a rapid change in electrical potential to a carbon-fibre electrode and measurement of induced

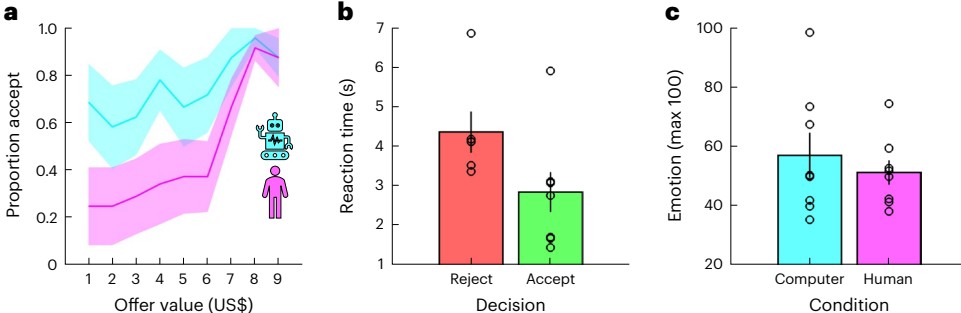

**Fig. 2 | Behavioural results. a**, Proportion accept (*y* axis) given offer value (*x* axis) and condition (colours). **b**, Reaction time (*y* axis) given choice (colours). **c**, Emotion (*y* axis; 0–100) given condition (colours). **b**,**c**, Each dot is a dataset. **a**–**c**, Data are represented as mean ± s.e.m. across datasets (*n* = 8).

electrochemical reactions as changes in current at the electrode tip. The current responses carry information about not only the identity but also the concentration of neuromodulators in the surrounding neural tissue. This information is extracted using a signal prediction model trained on large wet-lab datasets where the chemical environment can be carefully controlled (see Supplementary Fig. 1 for an illustration of the electrochemical approach as well as an in vitro evaluation of the signal prediction model; see Methods for details).

The clinical treatment involved two surgeries 14–28 days apart for the bilateral implantation of DBS electrodes in the subthalamic nucleus of each hemisphere, and participants performed the ultimatum game in each surgical session (4 patients × 2 sessions = 8 datasets). Within a session, participants played the role of the responder in a series of one-shot interactions with human or computer avatars (30 trials × 2 conditions = 60 trials; Fig. 1b). The human avatars were unique, with each avatar having its own name and visual image, whereas the computer avatar was the same on each trial. The two conditions were blocked within a session and counterbalanced across sessions. In each condition, participants were asked to accept or reject splits of a US$20 stake. The human and computer avatars were both programmed to make offers between US$1 and US$9, with the same set of offers used in each condition in a session but in a randomized order. Participants were not given this information. On around a third of trials, participants were asked to rate how they felt about the game by moving a slider along a visual analogue mood scale ranging from negative (sad emoji) to positive (happy emoji).

**Behavioural results**

To unpack participants' task behaviour, we first ran a logistic mixed-effects model in which we predicted choices (reject = 0, accept = 1) using offer value, condition (computer = −0.5, human = 0.5), and the interaction between these terms (choice model 1 (C-M1); see Methods for details about the statistical analysis of behavioural and neural data which was conducted at the trial level). As expected, value had a positive effect on choice (positive slopes in Fig. 2a; C-M1, value, $t(454) = 2.43$, $P = 0.016$, $\beta \pm 95\%$ confidence interval (CI) = 1.77 ± 1.43): participants accepted 43% of offers smaller than, or equal to, one standard deviation (US$2) below the mean (US$5), 58% of offers within one standard deviation of the mean, and 94% of offers equal to, or larger than, one standard deviation above the mean. In keeping with earlier results[30–33], condition had a negative effect on choice (pink below cyan in Fig. 2a; C-M1, condition, $t(454) = -2.23$, $P = 0.026$, $\beta \pm 95\%$ CI = −3.60 ± 3.18): participants accepted 50% of the offers made by human avatars but 75% of those made by the computer avatar. There was no interaction between value and condition (C-M1, value × condition, $t(454) = 1.35$, $P = 0.176$, $\beta \pm 95\%$ CI = 1.70 ± 2.47).

Previous studies have found that participants adapt their willingness to accept an offer to the history of offer values, including when offers are made by unique avatars and offer values are randomized[36,37]. To test for behavioural adaptation, we re-ran the logistic mixed-effects

model after adding the difference in value between the current and the previous offer—akin to a one-step RPE—as a predictor (C-M2). The effects of value and condition remained (C-M2; value, $t(423) = 2.82$, $P = 0.005$, $\beta \pm 95\%$ CI = 1.79 ± 1.25; condition, $t(423) = -2.38$; $P = 0.018$, $\beta \pm 95\%$ CI = −3.67 ± 3.03; value × condition, $t(423) = 0.90$, $P = 0.371$, $\beta \pm 95\%$ CI = 1.13 ± 2.47), but there were no history effects (C-M2; value difference, $t(423) = 0.38$, $P = 0.707$, $\beta \pm$ CI = 0.12 ± 0.61; value difference × condition, $t(423) = 0.59$, $P = 0.558$, $\beta \pm 95\%$ CI = 0.47 ± 1.57). Indicative of some behavioural adaptation being at play, the model that included history effects provided a better fit to the data than the basic model (Akaike information criterion (AIC), C-M1 = 3016, C-M2 = 2,843). We note that the absence of an interaction between value difference and condition indicates that the use of a single avatar (computer) versus multiple avatars (human) did not affect task behaviour.

We next used linear mixed-effects models to analyse reaction times (RT model 1 (RT-M1)) and the emotion ratings (emotion model 1 (E-M1)). In addition to the terms from the choice analysis (C-M1), we included choice (reject = −0.5, accept = 0.5) and the interactions between choice and the other terms. There was an effect of choice on reaction times, with participants taking longer to reject than accept an offer (Fig. 2b; RT-M1, choice, $t(450) = -2.56$, $P = 0.011$, $\beta \pm 95\%$ CI = −0.37 ± 0.28). However, there were no effects of value or condition on reaction times (RT-M1, all absolute $t(450) < 1.62$, all $P > 0.106$). In addition, the analysis of the emotion ratings did not return any effects (Fig. 2c; E-M1, all absolute $t(147) < 1.70$, all $P > 0.093$).

**Overall dopamine tracks social context**

Having characterized participants' task behaviour, we turned to the electrochemical data to test whether dopamine and serotonin tracked the social context and value signals embedded in our task. Considering the results obtained from pharmacological manipulations of dopamine and serotonin levels during economic games[10,11,14,30], we first asked whether overall levels of dopamine/serotonin varied with the decision to accept or reject an offer and the social context. To this end, we ran a linear mixed-effects model in which we predicted trial-by-trial estimates of overall levels of dopamine/serotonin using choice, condition and their interaction (neural model 1 (N-M1)). Here, overall dopamine/serotonin levels were defined as the sum of samples within a 1 s window after offer presentation.

This analysis indicated that overall levels of dopamine, but not serotonin, were modulated by social context. Specifically, while there were no choice-related effects on dopamine, there was a positive effect of condition, with higher dopamine in the human than the computer condition (pink above cyan in Fig. 3a, top; N-M1; choice, $t(454) = -0.29$, $P = 0.771$, $\beta \pm 95\%$ CI = −0.03 ± 0.20; condition, $t(454) = 3.04$, $P = 0.002$, $\beta \pm 95\%$ CI = 0.85 ± 0.55; choice × condition, $t(454) = -0.69$, $P = 0.493$, $\beta \pm 95\%$ CI = −0.17 ± 0.50). In other words, while dopamine may drive a general change in the willingness to accept an offer made by a human versus a computer (Fig. 2a), it does not drive individual choices per se. In contrast, there were no effects on serotonin (Fig. 3a, bottom; N-M1;

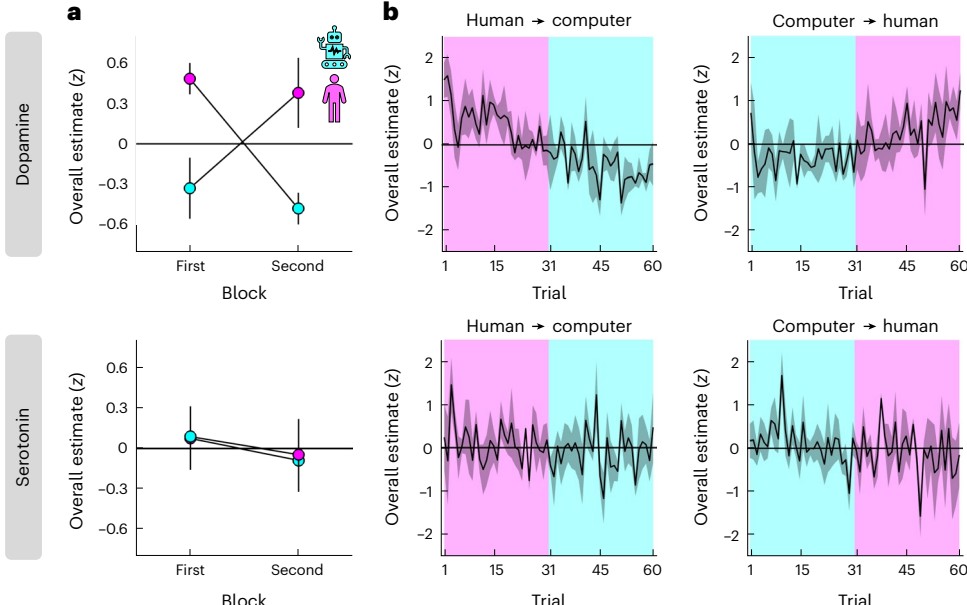

**Fig. 3 | Overall dopamine levels depend on social context. a**, Overall dopamine/serotonin (y axis) separated by condition (colour) and its order within a session (x axis). **b**, Overall dopamine/serotonin (y axis) across time separated by condition order (left column versus right column). **a,b**, Overall estimates were computed as the sum of neuromodulator samples within a 1 s window (ten samples) after offer presentation. We limited the estimates to this window for consistency with the relative analysis in Fig. 4 and to ensure that all estimates were based on the

same number of samples regardless of variation in reaction times and trial events (for example, variable duration of proposer screen and emotion ratings). The effect of condition on dopamine remained regardless of the specific time window (for example, a 6 s window centred on offer presentation; N-M1, $t(454) = 2.63$, $P = 0.009$, $\beta \pm 95\%$ CI $= 1.00 \pm 0.75$). Data are represented as mean $\pm$ s.e.m. across datasets ($n = 4$ for each order). Each dataset (session) was z-scored separately.

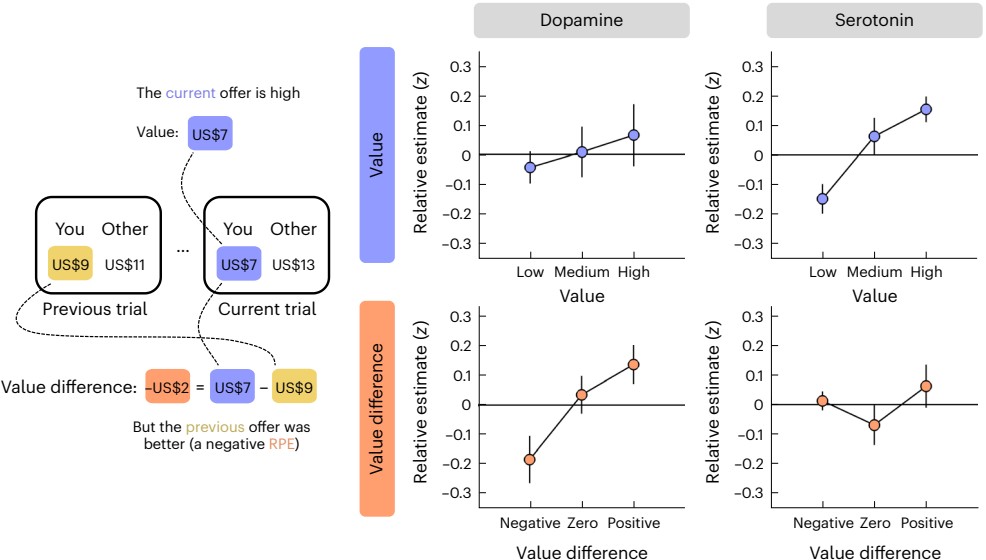

**Fig. 4 | Relative changes in dopamine and serotonin reflect distinct value signals.** To visualize the value effects (see main text for statistical analysis), we grouped the value of the current offer (top row) and the difference in value between the current and the previous offer (bottom row) into three bins using terciles and then plotted relative dopamine/serotonin (y axis) for each bin (x

axis). Relative estimates were computed by first subtracting the sample at offer presentation as a local baseline and then taking the sum of neuromodulator samples within a 1 s window (ten samples) after offer presentation. Data are represented as mean $\pm$ s.e.m. across datasets ($n = 8$). Each dataset (session) was z-scored separately.

choice, $t(454) = -0.57$, $P = 0.570$, $\beta \pm 95\%$ CI $= -0.06 \pm 0.21$; condition, $t(454) = -0.01$, $P = 0.993$, $\beta \pm 95\%$ CI $= 0.00 \pm 0.58$; choice × condition, $t(454) = -1.21$, $P = 0.227$, $\beta \pm 95\%$ CI $= -0.32 \pm 0.53$).

Our counterbalanced block design (Fig. 1b) allowed us to control for order effects such as changes in task engagement due to fatigue. To this end, we re-ran the linear mixed-effects model after including the order of a condition within a session (first = −0.5, second = 0.5) and its interactions with the other terms as predictors (N-M2).

In further support of an interpretation that overall dopamine levels depend on social context, the analysis again identified a positive effect of condition, but no order-related effects, on dopamine (compare order in Fig. 3a, top; N-M2; condition, $t(450) = 2.98$, $P = 0.003$, $\beta \pm 95\%$ CI $= 0.85 \pm 0.56$; remaining effects, all absolute $t(450) < 0.76$, all $P > 0.451$). There were again no effects on serotonin (Fig. 3a, bottom; N-M2; all absolute $t(450) < 1.37$, all $P > 0.172$). Consistent with an absence of order-related effects, the model that included order effects provided

a worse fit to the data compared to the basic model (AIC; dopamine, N-M1 = 1,205, N-M2 = 1,263; serotonin, N-M1 = 1,270, N-M2 = 1,331). Visualization of the full trial-by-trial data indicated that the effect of social context on overall dopamine levels reflected gradual changes across time (Fig. 3b).

### Relative dopamine and serotonin track value signals

Having established an impact of social context on overall dopamine levels, we asked whether changes in dopamine/serotonin relative to a local baseline reflected value signals as predicted by both empirical and theoretical work[15–19]. To this end, we ran a linear mixed-effects model in which we predicted trial-by-trial estimates of relative changes in dopamine/serotonin using the value of the current offer and the difference in value between the current and the previous offer—akin to a one-step RPE (N-M3). Here, relative dopamine/serotonin levels were defined as the sum of samples within a 1 s window after offer presentation, but now after having first subtracted the sample at offer presentation as a local baseline.

This analysis indicated that relative changes in dopamine and serotonin reflected distinct yet complementary value signals. In support of a role in RPE signalling, value difference, but not value per se, had a positive effect on dopamine (Fig. 4, left column; N-M3; value, $t(426) = -1.47$, $P = 0.144$, $\beta \pm 95\%$ CI = $-0.10 \pm 0.14$; value difference, $t(426) = 2.82$, $P = 0.005$, $\beta \pm 95\%$ CI = $0.21 \pm 0.14$): dopamine showed a relative decrease when the current offer was lower than the previous one (akin to a negative RPE) and a relative increase when the current offer was higher than the previous one (akin to a positive RPE). In contrast to this response pattern, value, but not value difference, had a positive effect on serotonin (Fig. 4, right column; N-M3; value, $t(426) = 3.15$, $P = 0.002$, $\beta \pm 95\%$ CI = $0.22 \pm 0.14$; value difference, $t(426) = -1.82$, $P = 0.070$, $\beta \pm 95\%$ CI = $0.13 \pm 0.14$): serotonin showed a relative increase for high offers and a relative decrease for low offers. To control for any model misestimation due to the correlation between value and value difference (Pearson's $r = 0.73$), we regressed value difference against value and used the residuals (Pearson's $r \approx 0$) as our predictor for value difference (N-M3*). In keeping with the original results, this analysis returned an effect of value difference on dopamine (N-M3*; value, $t(426) = 0.82$, $P = 0.412$, $\beta \pm 95\%$ CI = $0.05 \pm 0.11$; value difference, $t(426) = 2.82$, $P = 0.005$, $\beta \pm 95\%$ CI = $0.21 \pm 0.14$) and an effect of value on serotonin (N-M3*; value, $t(426) = 2.67$, $P = 0.008$, $\beta \pm 95\%$ CI = $0.13 \pm 0.09$; value difference, $t(426) = -1.82$, $P = 0.070$, $\beta \pm 95\%$ CI = $-0.13 \pm 0.14$).

Recent research suggests that serotonin tracks absolute (unsigned) RPEs, which provide an estimate of variability in the environment and can be used to regulate the rate of learning[19,38]. Since the visualization of relative changes in serotonin was consistent with this computational function (U shape in Fig. 4, bottom right), we tested this hypothesis formally by also including the absolute value difference as a predictor (N-M4). However, while this analysis replicated the original results, it did not identify an effect of absolute value difference for dopamine (NM-4; value, $t(425) = -1.48$, $P = 0.139$, $\beta \pm 95\%$ CI = $-0.11 \pm 0.14$; value difference, $t(425) = 2.87$, $P = 0.004$, $\beta \pm 95\%$ CI = $0.21 \pm 0.14$; absolute value difference, $t(425) = 0.92$, $P = 0.360$, $\beta \pm 95\%$ CI = $0.08 \pm 0.17$) or serotonin (N-M4; value, $t(425) = 3.07$, $P = 0.002$, $\beta \pm 95\%$ CI = $0.22 \pm 0.14$; value difference, $t(425) = -1.78$, $P = 0.076$, $\beta \pm 95\%$ CI = $-0.13 \pm 0.14$; absolute value difference, $t(426) = 0.18$, $P = 0.861$, $\beta \pm 95\%$ CI = $0.02 \pm 0.15$). In addition, the model itself provided a worse fit to the data compared to the basic model AIC; dopamine, N-M3 = 1,231, N-M4 = 1,243; serotonin, N-M3 = 1,231, N-M4 = 1,244).

Finally, we asked whether the value-related effects were modulated by social context. To this end, we re-ran the linear mixed-effects model after including condition and its interaction with the value-related terms as predictors (N-M5). In support of a hypothesis that relative changes in dopamine and serotonin reflect generalized value signals, the analysis replicated the earlier effects, but did not identify any

condition-related effects, for dopamine (N-M5; value, $t(423) = -1.36$, $P = 0.176$, $\beta \pm 95\%$ CI = $-0.11 \pm 0.16$; value difference, $t(423) = 2.63$, $P = 0.009$, $\beta \pm 95\%$ CI = $0.21 \pm 0.15$; condition-related effects, all absolute $t(423) < 0.93$, all $P > 0.354$) or serotonin (N-M5; value, $t(423) = 3.00$, $P = 0.003$, $\beta \pm 95\%$ CI = $0.21 \pm 0.14$; value difference, $t(423) = -1.72$, $P = 0.086$, $\beta \pm 95\%$ CI = $-0.12 \pm 0.14$; condition-related effects, all absolute $t(423) < 0.86$, all $P > 0.390$). In line with an absence of modulation by social context, the model itself provided a worse fit to the data compared to the basic model (AIC; dopamine, N-M3 = 1,231, N-M5 = 1,262; serotonin, N-M3 = 1,231, N-M5 = 1,273).

## Discussion

Previous work suggests that dopamine and serotonin play central roles in human social interaction[9,39]. However, because of methodological limitations, the contribution of these neuromodulators to social behaviour has not yet been studied at fast timescales in humans. By applying a recently developed method for human electrochemistry during DBS surgery[22–26], we obtained subsecond estimates of dopamine and serotonin from the SNr while patients with Parkinson's disease played the ultimatum game with both human and computer avatars. Despite receiving the same offers in both conditions, participants rejected more human than computer offers, indicative of the human condition invoking social fairness norms. The electrochemical data indicated that dopamine underpinned this behavioural response, with higher overall levels of dopamine, but not serotonin, in the human condition. Regardless of social context, and in support of a hypothesis that dopamine and serotonin carry distinct yet complementary value signals, changes in dopamine relative to a local baseline tracked trial-by-trial changes in offer value, whereas relative changes in serotonin tracked the current offer value. Taken together, these results indicate that dopamine and serotonin support not only the computation of value statistics but also the norm-based use of these statistics during social interaction.

Our behavioural data replicated the result that people reject more offers when they believe they are interacting with another person as opposed to a computer[30–33]. This effect of social context, which is accompanied by increased affective arousal as measured by skin conductance[32] and increased activity in emotion-related brain regions (for example, amygdala, insula and striatum)[30,33], has been attributed to human social interaction invoking a sense of fairness. While enforcing fairness norms can promote cooperation[40], research suggests that our sense of fairness is in fact self-oriented: we view unfair offers as displays of dominance and reject them to avoid the imposition of inferior status[41] or gain social control[42]. Such a change in the frame of reference for social interaction may explain why overall dopamine levels were higher for human than computer avatars. Indeed, pharmacological studies have found that elevated dopamine levels make people more averse to differences between their own and others' payoffs[43], less averse to inflicting pain on others in exchange for money[14] and more selfish when selfish behaviours cannot be punished[44]. One prediction of the hypothesis that dopamine helps set the stage for social interaction is that disturbances in dopamine signalling should increase the risk of social dysfunction. Indeed, schizophrenia, associated with a dysregulated dopamine system[45], can involve delusions centred around social themes (for example, persecutory delusions)[46,47], sometimes ruining people's social lives. The hypothesis also fits with a growing literature linking dopamine to biases in social reasoning[48–50], such as attribution of harmful intent. While pharmacological studies have found a link between overall serotonin levels and the willingness to accept unfair offers[10,11], dietary acute tryptophan depletion does not influence neural discrimination between human and computer conditions in the ultimatum game as assessed by fMRI[30]. In line with this result, we found no effect of social context on overall serotonin levels.

In addition to overall levels, we investigated how dopamine and serotonin changed relative to a local baseline, here the presentation of the current offer. Consistent with the RPE theory of dopamine[15,16],

we found that relative changes in dopamine reflected the difference in value between the current and the previous offer: dopamine showed a relative decrease when value decreased (a negative RPE) and a relative increase when value increased (a positive RPE) regardless of the social context. This result from the human SNr fits with previous animal work, which found that the activity of SNr neurons is indicative of modulation by RPEs[51]. In contrast, relative changes in serotonin reflected the value of the current offer, with a relative decrease for low values and a relative increase for high values regardless of the previous offer and the social context. Taken together, these response patterns indicate that dopamine and serotonin play complementary rather than opponent roles in value-based processes[52,53]—with dopamine supporting a comparison of the present with the past and serotonin supporting an evaluation of the here and now—and that these roles generalize across contexts.

Our electrochemical data were collected from the SNr (Fig. 1a); we should therefore consider (1) its anatomical connections and (2) whether our results are specific to the SNr or reflect signals that are broadcasted widely within the brain. First, the SNr is one of the basal ganglia's main output nuclei: it receives excitatory glutamatergic inputs from the subthalamic nucleus[54], inhibitory GABAergic inputs from the striatum[54], dopaminergic inputs from the SNc[55] and serotonergic inputs from the raphe nucleus[56,57]; then, it sends GABAergic outputs to the thalamus[54], which control glutamatergic outputs from the thalamus—a main relay station for sensorimotor information—to cortical and subcortical regions[54,58–61]. These distal projection targets include regions that support decision-making in non-social and social contexts, including the orbitofrontal cortex, the medial prefrontal cortex, the anterior cingulate cortex and the amygdala[4,5]. Second, it is hard to say whether our results are specific to the SNr. Dopamine release in the SNr is mainly driven by somatodendritic release from the SNc[55], but this mechanism can be activated by action potentials in the SNc that drive synaptic release in other brain regions[62]. Similarly, while serotonin release in the SNr is mainly driven by direct synaptic release from the raphe nucleus[56,57], the upstream serotonergic neurons may project to other brain regions. Future research could address regional specificity by recording from multiple brain regions on the same task—with a greater diversity in recording targets provided by the recent extension of human electrochemistry to depth electrodes implanted throughout the brain for epilepsy monitoring[63].

Our electrochemical data necessarily had to be collected in brain surgery patients; in our case, Parkinson's disease patients undergoing bilateral DBS surgery. While Parkinson's disease is characterized by a loss of midbrain dopamine neurons[64], there are several reasons why our results are likely to generalize to the 'healthy' brain. First, the patients' disease progression was not so severe that DBS would have been unlikely to be effective (Supplementary Table 1). Second, even though the patients may have a general reduction in dopamine levels, this reduction cannot explain the difference in overall dopamine levels between conditions in our within-subject design. Third, indicative of an otherwise normal range of brain function, the patients did not present with notable cognitive impairment or refractory psychiatric disorders (Supplementary Table 1), both contra-indications for DBS. Fourth, our results are unlikely to be confounded by medication considering that Parkinson medication was withheld during surgery and that the patients otherwise received different medications (Supplementary Table 1). Fifth, previous studies applying human electrochemistry during DBS surgery have seen comparable dopamine and serotonin responses in Parkinson's disease and essential tremor[25], with the latter condition involving small, or no, disturbances in the dopamine and serotonin systems[65]. Finally, the value-related results for dopamine are consistent with a large body of animal work on RPE signalling in the basal ganglia[66].

Given the novelty of human electrochemistry, we decided to use a simple, widely used social task. The one-shot version of the ultimatum game has, for example, been used to study cross-cultural variation in social fairness norms[13,67], the neural basis of social behaviour[33], the social impact of brain injury[36,68] and pharmacological manipulations of neuromodulators[10,30] or hormones, such as oxytocin[69] and testosterone[70,71]. Similarly, the human versus computer manipulation of social context has been used to assess social specificity beyond the ultimatum game[30,33] in a range of social neuroscience studies[72,73]. However, a complete understanding of the role of fast dopamine and serotonin signalling in human social behaviour requires future experiments that involve repeated interaction and sophisticated inference, such as multi-round economic games[74–76]. In conclusion, our study provides direct evidence from the human brain that fast changes in dopamine and serotonin reflect context and value signals during social interaction, and that the distinct yet complementary roles of dopamine and serotonin in value coding generalize across contexts.

## Methods

### Ethics
The study complies with all relevant ethical regulations and was approved by the Institutional Review Boards at the Icahn School of Medicine at Mount Sinai (13-00415) and Virginia Tech (11-078). No adverse or unanticipated events occurred during or as a result of the study.

### Participants
Four Parkinson's disease patients (one female, mean age ± s.e.m. = 71.3 ± 3.4 years) participated in the study. Once they had agreed to the DBS treatment, they were assessed for suitability for the research study and given the option to participate. Before obtaining informed written consent, the research team provided both written and verbal information about the research study and how it would alter the clinical procedure. Specifically, patients were informed that the study would involve a research-exclusive probe (carbon-fibre electrode) and that extra time (maximum 30 min) would be needed to complete the study. Patients did not receive compensation for participation, and they knew that they would not receive any money earned in the task.

### Behavioural testing
**Surgical sessions.** Each participant performed the task in two surgical sessions for the bilateral implantation of DBS electrodes in the subthalamic nucleus of each hemisphere. The sessions were 14–28 days apart. Participants performed a practice version of the task before and during surgery. During surgery, participants laid in a semi-upright position and viewed a computer monitor at a distance of around 100 cm. Participants used a gamepad to submit their responses.

**Ultimatum game.** Participants performed the ultimatum game, a two-person 'take-it-or-leave-it' game probing social fairness norms[12]. The task was implemented using PsychoPy[77]. Participants played the role of the responder in a series of one-shot interactions with a human or a computer avatar as the proposer. The human avatars were unique, with each avatar having its own image and name, whereas the computer avatar was the same on every trial. The human avatars were designed to cover a diverse range of racial and cultural identities. Participants experienced two different sets of human avatars in the two sessions. The human versus computer conditions were blocked within a session (2 conditions × 30 trials = 60 trials in a session) and were counterbalanced across sessions.

On each trial, participants were first shown an avatar indicating the current proposer (1.8–2.3 s). They were then shown the proposed split of a US$20 stake and had to decide whether to accept or reject the offer (self-paced). Unbeknownst to participants, the human and computer avatars were preprogrammed to make offers in the range of US$1–9. The same set of offers were used in the human and computer conditions within a given session but in a randomized order. If participants accepted the offer, both parties would receive the proposed amounts.

If they rejected the offer, both parties would get nothing. Once participants had made their decision, feedback screens first highlighted the chosen option (1 s) and then the outcome of the decision (1 s). Participants were then shown a blank screen (1 s), before continuing to the next trial. However, on around a third of the trials, participants were first asked to indicate how they felt about the game (self-paced) by moving a slider along a visual analogue mood scale ranging from negative (sad emoji) to positive (happy emoji).

## Statistical analysis

**Trial exclusions.** We excluded trials where the reaction time was longer than 14 s to minimize the impact of any momentary distraction in the operating room (~5% of trials). When analysing history effects, we excluded trials preceded by an already excluded trial and the first trial of a block.

**Mixed-effects models.** We used mixed-effects models specified at the trial level for statistical analysis of behavioural and neural data. All models included (1) fixed (population-level) effects and (2) random effects varying by dataset (session) with a free covariance matrix. We note that removing all random effects except for the intercept from a particular model did not change the significance of any of the reported effects for that model. We used the AIC for model comparison when relevant. All statistical tests are two-tailed. The mixed-effects models were implemented using the 'fitglme' function in MATLAB (MathWorks).

**Behavioural models.** In Wilkinson notation, the logistic choice models (C; reject = 0, choice = 1) were specified as:

 C-M1, C ~ 1 + value × condition + (1 + value × condition | dataset)

 C-M2, C ~ 1 + condition × (value + value difference) + (1 + condition × (value + value difference) | dataset)

 The linear reaction time model (RT) was specified as:

 RT-M1, RT ~ 1 + choice × value × condition + (1 + choice × value × condition | dataset)

 The linear emotion rating model (E) was specified as:

 E-M1, E ~ 1 + choice × value × condition + (1 + choice × value × condition | dataset)

**Neural models.** In Wilkinson notation, the linear neural models (N) were specified as:

 N-M1, overall N ~ 1 + condition × choice + (1 + condition × choice | dataset)

 N-M2, overall N ~ 1 + condition × choice × order + (1 + condition × choice × order | dataset)

 N-M3, relative N ~ 1 + value + value difference + (1 + value + value difference | dataset)

 N-M4, relative N ~ 1 + value + value difference + absolute value difference + (1 + value + value difference + absolute value difference | dataset)

 N-M5, relative N ~ 1 + condition × (value + value difference) + (1 + condition × (value + value difference) | dataset)

 For N-M3*, which controls for correlations between value and value difference, we regressed value difference against value and used the residuals as our predictor for value difference.

**Coding and standardization of variables.** Binary variables were contrast coded (−0.5 or 0.5) and continuous variables were standardized separately for each dataset using a $z$-score transformation. We performed standardization, which transforms data into a relative frame of reference, for several reasons. First, it facilitates the comparison of fitted coefficients within a given model. For example, while value difference is derived from value, their fitted coefficients cannot be compared without standardization as the raw variables have different means and variances. Second, it facilitates the comparison of fitted coefficients when the same model is applied to different data. For example, if value

had been found to have a positive effect on both relative dopamine and serotonin, then the fitted coefficients could be compared across the neuromodulators. Third, in the case of neural data, standardization minimizes, if not removes, the influence of any unmodelled sources of dataset-level variation in the baseline and/or the variance of the data. Finally, standardization mitigates against between-dataset differences due to trial exclusions, which again can affect the mean and/or the variance of the data.

**Neuromodulator estimates.** The 'overall' estimates were computed as the sum of samples within a 1 s window (ten samples) after offer presentation. The 'relative' estimates were computed by first subtracting the sample at offer presentation as a local baseline and then taking the sum of neuromodulator samples within a 1 s window (ten samples) after offer presentation. We computed the overall and relative estimates using the same time window for consistency and to ensure that the overall estimates were based on the same number of samples regardless of reaction times and trial events (for example, variable duration of proposer screen and emotion ratings).

## Electrochemistry

Here, we first provide a general description of our approach, before detailing its implementation in the current study.

**General description.** Human electrochemistry[22–26] builds on fast-scan cyclic voltammetry (FSCV) as used in animal work[78,79]. The carbon-fibre electrodes are made in the same way as those used in rodents[34], except with dimensions modified for use in the human brain[22]. The data acquisition protocol is similar to that used in rodents with regards to the time course of the voltage sweeps and the recording of the induced current time series during those sweeps[35]. The main change from animal work is the statistical method used to estimate the concentration of analytes of interest from the measured current time series (Supplementary Fig. 1).

In brief, FSCV involves the delivery of a rapid change in electrical potential to an electrode and measurement of the induced electrochemical reactions as changes in current at the electrode tip, with the guiding idea being that the current response carries information about both the identity and the concentration of analytes in the surrounding neural tissue. The goal of an analysis method for FSCV data is therefore to develop a statistical model that uses the current response in the best possible way to separate and estimate analytes of interest. The standard procedure is to train the statistical model on in vitro data collected in the laboratory where the presence and concentration of analytes of interest can be controlled and then apply this model to in vivo data for signal prediction.

Traditionally, the statistical model involves a decomposition of the in vitro training data into principal components that are then used for in vivo analyte inference within a regression framework[80]. In broad terms, this approach treats analyte inference as a problem of signal reconstruction: the concentration of an analyte of interest is estimated by mapping an in vivo current response onto those collected in vitro and then using the best match to label the in vivo current response. We instead treat analyte inference as a problem of signal prediction, with the statistical model optimized to generate accurate predictions about out-of-training data. Previous human work[23–26] has used elastic net regression[81], but recent years have seen the development of more powerful machine-learning methods[63]. Here, we used deep convolutional neural networks. Since information is distributed throughout a current time series, and not only at the oxidation or reduction peaks revealed by principal components analysis[23–25], we use non-decomposed data such that every time point within a current time series contributes to signal prediction. To facilitate and evaluate out-of-training prediction, we train the model using large in vitro datasets and evaluate it using cross-validation.

There are several statistical advantages to this approach to analyte inference. First, objective classification sidesteps the need for experimenter judgement (for example, deciding on the cut-off for the

number of principal components based on reconstructed variance and visual inspection of background-subtracted voltammograms). Second, reframing analyte inference as a problem of signal prediction means that the statistical model can be directly evaluated using in vitro data that were withheld from training. Third, such out-of-training evaluation can reveal whether there is any bias in the assembly of training data or overfitting to the training data.

Earlier work has taken steps to validate human electrochemistry. First, the human-compatible carbon-fibre electrodes have similar electrochemical properties to those used in rodents[22]. Second, the signal prediction approach returns more reliable neuromodulator estimates than principal component regression[23]. Third, it does not confuse changes in pH for changes in neuromodulators[23–25]. Fourth, it does not confuse neuromodulators with one another[24–26,63]. Fifth, it returns accurate neuromodulator estimates when tested in a laboratory setting where two neuromodulators simultaneously change across time[25].

**Carbon-fibre electrodes.** As part of the surgical procedure to implant a DBS electrode, we made electrochemical recordings of dopamine and serotonin fluctuations using a carbon-fibre electrode. The carbon-fibre electrode was temporarily inserted into the SNr along a guide cannula positioned in accordance with DBS planning. The carbon-fibre electrode was modified to fit 24.5 cm Alpha Omega NeuroProbe Sonus Guide Tubes (STR-901080-10). Electrode construction and the mobile electrochemical recording station are described in detail in previous work[22,23].

**Data acquisition protocol.** Our FSCV protocol was based on earlier work in both rodents[34,35] and humans[22–26] and implemented using pCLAMP (Axon Instruments). Our measurement waveform was a standard triangular voltage waveform (ramp up from $-0.6$ V to $+1.4$ V at $400$ V s$^{-1}$, ramp down from $+1.4$ V to $-0.6$ V at $-400$ V s$^{-1}$). While patients were being prepared to perform the task, we ran a conditioning protocol consisting of a 97 Hz application of the measurement waveform (hold at $-0.6$ V for 0.32 ms, ramp up to $+1.4$ V at 400 V s$^{-1}$, ramp down to $-0.6$ V at $-400$ V s$^{-1}$, and repeat) to allow equilibration of the recording surface. Then, during the task, a 10 Hz application of the measurement waveform was applied for the entire duration of the experiment (hold at $-0.6$ V for 90 ms, ramp up to $+1.4$ V at 400 V s$^{-1}$, ramp down to $-0.6$ V at $-400$ V s$^{-1}$, and repeat) with a base 100 kHz sampling rate.

**Signal prediction model.** We generated in vivo signal predictions using an ensemble of deep convolutional neural networks that were trained and cross-validated on in vitro data with known concentrations of dopamine, serotonin, norepinephrine and pH. The model architecture was based on the InceptionTime time series classification model[82] but modified for a regression framework. The model was implemented in Python[83] using TensorFlow[84] and Keras[85]. Following previous applications[82], equally weighted averages of in vivo signal predictions from multiple InceptionTime models were used to account for variability in the training process.

Here, the InceptionTime model is based on two residual neural network (ResNet)[86] blocks, each containing three convolutional blocks. The input is added to the output of the first ResNet block, and, after activation, this serves as the input to the next ResNet block. It is also added to the output of the second ResNet block, and the sum flows through activation functions, global average pooling and then finally to a dense layer with four output nodes, which, after activation, give the predictions of dopamine, serotonin, norepinephrine and pH. Each of the three convolutional layers in a ResNet block is composed of convolutional blocks having four convolutional layers with 32 filters and increasing kernel sizes (1, 10, 20 and 40). The output of each of these convolutional layers is stacked together, after which batch normalization and activation are applied. This output serves as the input for the next convolutional block. All activation functions are rectified linear units, except after the last dense layer, which uses softplus.

**Training data.** In vitro training data consisted of 64 datasets collected by exposing 64 carbon-fibre electrodes to varying concentrations (0–2,500 nM at -7.4 pH) of dopamine, serotonin and norepinephrine in 0.5 M phosphate buffered saline. In addition to mono-analyte solutions, each dataset included mixture solutions and pH solutions in which the pH ranged from 7.0 to 7.8 while the neuromodulators were held at 0 nM. We randomized the order of solutions for each analyte within a dataset and the order of analytes across datasets. To collect the data, the carbon-fibre electrode was laid horizontally in a flow cell and solutions were added to the flow cell using a 5 ml syringe on a solution-by-solution basis. The carbon-fibre electrode was precycled at 97 Hz for 27 s before each 65 s, 10 Hz data collection. Currents measured from the most stable 15 s epoch (150 sweeps) were used for model training; this step was taken to reduce variation caused by electrical noise and equilibration. Of the 64 datasets, five were held out of training for testing purposes and the remaining 59 were used for training. This resulted in a training set consisting of 7,260 unique concentration combinations and 1,089,000 current sweeps. The test set contained 795 concentration combinations and 119,250 current sweeps.

**Model training and evaluation.** We took the first differential of the current sweeps for each concentration across the training datasets. These differentiated current sweeps were normalized ($z$-score) and shifted up by ten standard deviations to avoid zero gradients. Ninety per cent of these data were used for model training while the remaining data were used for model validation. The data were split such that data from a given probe within a given concentration were in the same set. The model was trained using the Adam optimizer[87] as implemented in TensorFlow, with an initial learning rate of $1 \times 10^{-3}$, mean squared error loss and a batch size of 64. After each epoch, the loss on the validation set was calculated. If the performance on the validation set did not improve for five consecutive epochs, then the learning rate was halved until it reached a minimum of $1 \times 10^{-5}$. The model from the epoch with the lowest validation loss after 35 epochs was selected as the final model. The final model consisted of an ensemble of five equally weighted submodels from five separate training runs. The training process is non-deterministic as variability is introduced by stochastic gradient descent, the initialization of the initial weights and the order in which data are fed to the algorithm during training. Note that different training and validation sets were used for each training run to avoid model overfitting.

## Generating signal predictions

In vitro signal predictions for the five held-out training datasets and in vivo signal predictions for the datasets collected in the SNr were generated in the same way. In both cases, differentiated current sweeps were fed to the model and the mean estimate across the five submodels was used to generate concentration estimates for each current sweep after applying an inverse normalization procedure. The signal prediction model was evaluated by comparing predicted and labelled concentrations using the held-out training datasets. This step demonstrated that the signal prediction model had high sensitivity and high specificity (Supplementary Fig. 1).

## Reporting summary

Further information on research design is available in the Nature Portfolio Reporting Summary linked to this article.

## Data availability

Behavioural and neural data are available on GitHub: https://github.com/danbang/article-DA-5HT-UG-SNr.

## Code availability

Code for reproducing figures is available on GitHub: https://github.com/danbang/article-DA-5HT-UG-SNr.

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

## Acknowledgements

We thank the patient volunteers and the clinical staff at the Icahn School of Medicine at Mount Sinai for invaluable support and cooperation. This work was supported by the Lundbeck Foundation (D.B., grant no. R368-2021-325), Wellcome (D.B., grant no. 213630/Z/18/Z; P.R.M., grant no. 091188/Z/10/Z), the Swartz Foundation (P.R.M., grant no. 2019-11), NIH-NCATS (K.T.K., grant no. KL2TR001421), NIH-NIDA (K.T.K., grant no. R01-DA048096), NIH-NINDS (K.T.K. and P.R.M., grant no. R01-NS092701), NIH-NIMH (K.T.K., grant no. R01-MH121099; X.G., grant nos. R21-MH120789, R01-MH122611 and R01-MH123069; K.T.K., X.G. and P.R.M., grant no. R01-MH124115; P.R.M., grant nos. R01-MH122512 and R01-MH122948), a Virginia Tech Foundation Seale Innovation Award (P.R.M., grant no. FY22), seed funding from the Icahn School of Medicine at Mount Sinai (X.G.), the Max Planck Society (P.D.) and the Humboldt Foundation (P.D.). The Wellcome Centre for Human Neuroimaging is supported by core funding from Wellcome (grant no. 203147/Z/16/Z). The funders had no role in the study design, data collection and analysis, decision to publish or preparation of the manuscript. For the purpose of Open

Access, the authors have applied a CC BY public copyright licence to any Author Accepted Manuscript version arising from this submission.

## Author contributions

S.R.B., D.B., B.H.K., K.T.K., X.G. and P.R.M. conceptualized the project. B.H.K., K.Z., A.H., A.W.C., M.F. and H.S.M. were responsible for the patients. S.R.B. was responsible for the electrodes. B.H.K. performed the surgery. S.R.B., B.H.K., A.N.D., M.H., Q.F., O.P., J.P.W. and X.G. collected the data. Signal processing was completed by S.R.B., D.B., L.S.B., T.T., T.L., J.P.W., K.T.K. and P.R.M. The data were analysed by S.R.B. and D.B. Visualization was performed by S.R.B. and D.B. S.R.B., D.B., B.H.K., A.N.D., I.S., T.L., P.D., A.W.C., M.F., H.S.M., K.T.K., X.G. and P.R.M. interpreted the data. X.G. and P.R.M. supervised the project. The original draft was written by S.R.B. and D.B. The article was reviewed and edited by S.R.B., D.B., B.H.K., T.T., T.L., P.D., M.F., H.S.M., K.T.K., X.G. and P.R.M.

## Competing interests

The authors declare no competing interests.

## Additional information

**Correspondence and requests for materials** should be addressed to Seth R. Batten, Dan Bang, Xiaosi Gu or P. Read Montague.

# Reporting Summary

## Statistics

For all statistical analyses, confirm that the following items are present in the figure legend, table legend, main text, or Methods section.

| n/a | Confirmed | |
|---|---|---|
| ☐ | ☒ | The exact sample size (*n*) for each experimental group/condition, given as a discrete number and unit of measurement |
| ☐ | ☒ | A statement on whether measurements were taken from distinct samples or whether the same sample was measured repeatedly |
| ☐ | ☒ | The statistical test(s) used AND whether they are one- or two-sided *Only common tests should be described solely by name; describe more complex techniques in the Methods section.* |
| ☐ | ☒ | A description of all covariates tested |
| ☒ | ☐ | A description of any assumptions or corrections, such as tests of normality and adjustment for multiple comparisons |
| ☐ | ☒ | A full description of the statistical parameters including central tendency (e.g. means) or other basic estimates (e.g. regression coefficient) AND variation (e.g. standard deviation) or associated estimates of uncertainty (e.g. confidence intervals) |
| ☐ | ☒ | For null hypothesis testing, the test statistic (e.g. *F*, *t*, *r*) with confidence intervals, effect sizes, degrees of freedom and *P* value noted *Give P values as exact values whenever suitable.* |
| ☒ | ☐ | For Bayesian analysis, information on the choice of priors and Markov chain Monte Carlo settings |
| ☐ | ☒ | For hierarchical and complex designs, identification of the appropriate level for tests and full reporting of outcomes |
| ☐ | ☒ | Estimates of effect sizes (e.g. Cohen's *d*, Pearson's *r*), indicating how they were calculated |

*Our web collection on statistics for biologists contains articles on many of the points above.*

## Software and code

Policy information about availability of computer code

| Data collection | The experimental task was programmed in PsychoPy 2021.1.4. |
|---|---|
| | The behavioral data were recorded using PsychoPy 2021.1.4. |
| | The neural data were recorded using pCLAMP 10 Axon Instruments. |
| Data analysis | Dopamine and serotonin signal predictions were generated using a custom implementation of the InceptionTime time series classification model (Fawaz et al., Data Mining and Knowledge Discovery, 2020) in Python 3.9.7 using TensorFlow 2.6.0 and Keras 2.6.0. |
| | Behavioral and neural data were analyzed using standard statistical tests as implemented in MATLAB R2015b and MATLAB R2023a. |
| | Code for reproducing figres is available on GitHub: https://github.com/danbang/article-DA-5HT-UG-SNr. |

For manuscripts utilizing custom algorithms or software that are central to the research but not yet described in published literature, software must be made available to editors and reviewers. We strongly encourage code deposition in a community repository (e.g. GitHub). See the Nature Portfolio guidelines for submitting code & software for further information.

## Data

Policy information about availability of data

All manuscripts must include a data availability statement. This statement should provide the following information, where applicable:

- Accession codes, unique identifiers, or web links for publicly available datasets
- A description of any restrictions on data availability
- For clinical datasets or third party data, please ensure that the statement adheres to our policy

Behavioral and neural data are available on GitHub: https://github.com/danbang/article-DA-5HT-UG-SNr.

## Research involving human participants, their data, or biological material

Policy information about studies with human participants or human data. See also policy information about sex, gender (identity/presentation), and sexual orientation and race, ethnicity and racism.

| | |
|---|---|
| Reporting on sex and gender | We report the sex, age and disease of each participant. All participants have the same disease (Parkinson's disease). Sex and age were not included as covariates in any analysis. Details are provided in Supplementary Table 1. |
| Reporting on race, ethnicity, or other socially relevant groupings | We report the sex, age and disease status of each participant. All participants have the same disease (Parkinson's disease). Sex and age were not included as covariates in any analysis. Details are provided in Supplementary Table 1. |
| Population characteristics | Participants (n = 4, 1 female, mean age +/- SE = 71.3 +/- 3.4 years) were Parkinson's disease patients who underwent awake brain surgeries for the bilateral implantation of DBS electrodes in the subthalamic nucleus of each hemisphere. |
| Recruitment | Participants were recruited among Parkinson's disease patients who were scheduled to undergo awake brain surgeries for the bilateral implantation of DBS electrodes in the subthalamic nucleus of each hemisphere. Once they had agreed to the clinical treatment, they were assessed for suitability for the research study and given the option to participate. Before obtaining informed written consent, the research team provided both written and verbal information about the research study and how it would alter the clinical procedure. Specifically, patients were informed that the study would involve a research-exclusive probe (carbon-fiber electrode) and that extra time (maximum 30 min) would be needed to complete the study. This information was provided both verbally and in a written document. Patients did not receive compensation for participation, and they knew that they would not receive any money earned in the task. The recruitment procedure may have selected for patients who are very keen to contribute to science. |
| Ethics oversight | The study was approved by the IRB committees at the Icahn School of Medicine at Mount Sinai (13-00415) and Virginia Tech (11-078). |

Note that full information on the approval of the study protocol must also be provided in the manuscript.

# Field-specific reporting

Please select the one below that is the best fit for your research. If you are not sure, read the appropriate sections before making your selection.

☒ Life sciences          ☐ Behavioural & social sciences          ☐ Ecological, evolutionary & environmental sciences

For a reference copy of the document with all sections, see nature.com/documents/nr-reporting-summary-flat.pdf

# Life sciences study design

All studies must disclose on these points even when the disclosure is negative.

| | |
|---|---|
| Sample size | The opportunity to perform human electrochemistry as part of DBS surgery is rare. We collected the maximum number of participants possible within the study period. The sample size is in line with previous studies performing human electrochemistry as part of DBS surgery (e.g., Bang et al., Neuron, 2020). |
| Data exclusions | We excluded trials where the choice reaction time was longer than 14 s to minimize the impact of any momentary distraction in the operating room (around 5% of trials). In analyses of history effects, we excluded trials preceded by an already excluded trial and the first trial of a block. |
| Replication | The experimental findings were evaluated using statistical testing and associated test statistics and not through replication. |
| Randomization | Each participant performed the ultimatum game twice in two separate surgical sessions. For each participant, the human versus computer conditions were blocked within a session (2 conditions x 30 trials = 60 trials in a session) and counterbalanced across sessions. The order of the human versus computer conditions in the first session was counterbalanced across participants. |
| Blinding | Blinding was not relevant for the current study as awareness of the experimental conditions was a critical part of the study design. |

# Reporting for specific materials, systems and methods

We require information from authors about some types of materials, experimental systems and methods used in many studies. Here, indicate whether each material, system or method listed is relevant to your study. If you are not sure if a list item applies to your research, read the appropriate section before selecting a response.

| Materials & experimental systems | | Methods | |
|---|---|---|---|
| **n/a** | **Involved in the study** | **n/a** | **Involved in the study** |
| ☒ ☐ | Antibodies | ☒ ☐ | ChIP-seq |
| ☒ ☐ | Eukaryotic cell lines | ☒ ☐ | Flow cytometry |
| ☒ ☐ | Palaeontology and archaeology | ☒ ☐ | MRI-based neuroimaging |
| ☒ ☐ | Animals and other organisms | | |
| ☒ ☐ | Clinical data | | |
| ☒ ☐ | Dual use research of concern | | |
| ☒ ☐ | Plants | | |

