## [Peer Review File · Nature Human Behaviour]

Peer Review Information

Journal: Nature Human Behaviour

Manuscript Title: Dopamine and serotonin in human substantia nigra track social context and value signals during economic exchange

Corresponding author name(s): Seth R. Batten, Dan Bang, Xiaosi Gu, & P. Read Montague

Reviewer Comments & Decisions:

Decision Letter, initial version:

7th July 2023

Dear Dr Bang,

Thank you once again for your manuscript, entitled "Dopamine and serotonin in human substantia nigra track social context and value signals during economic exchange," and for your patience during the peer review process.

Your manuscript has now been evaluated by 3 reviewers, whose comments are included at the end of this letter. Although the reviewers find your work to be of interest, they also raise some important concerns. We are very interested in the possibility of publishing your study in Nature Human Behaviour, but would like to consider your response to these concerns in the form of a revised manuscript before we make a decision on publication.

To guide the scope of the revisions, the editors discuss the referee reports in detail within the team, including with the chief editor, with a view to (1) identifying key priorities that should be addressed in revision and (2) overruling referee requests that are deemed beyond the scope of the current study. We hope that you will find the prioritized set of referee points to be useful when revising your study. Please do not hesitate to get in touch if you would like to discuss these issues further.

In preparing a revised manuscript, we ask that you [1] address in full Reviewer #2's concerns by providing a convincing set of evidence in support of the chosen approach and an additional figure illustrating how the raw signals from the recordings are transformed and summarized; [2] substantially improve the description of the methods (more detail on the statistical modeling approach and the analytical choices, as requested by Reviewer #1); [3] discuss the specificity of the findings to the substantia nigra in comparison to other brain regions more classically associated with reward prediction errors (see comment by Reviewer #3).

In sum, we invite you to revise your manuscript taking into account all reviewer and editor comments. We are committed to providing a fair and constructive peer-review process. Do not hesitate to contact us if there are specific requests from the reviewers that you believe are technically impossible or unlikely to yield a meaningful outcome.

We hope to receive your revised manuscript within two months. I would be grateful if you could contact us as soon as possible if you foresee difficulties with meeting this target resubmission date.

- Include a "Response to the editors and reviewers" document detailing, point-by-point, how you addressed each editor and referee comment. If no action was taken to address a point, you must provide a compelling argument. When formatting this document, please respond to each reviewer comment individually, including the full text of the reviewer comment verbatim followed by your response to the individual point. This response will be used by the editors to evaluate your revision and sent back to the reviewers along with the revised manuscript.
- Highlight all changes made to your manuscript or provide us with a version that tracks changes.

[REDACTED]

We look forward to seeing the revised manuscript and thank you for the opportunity to review your work. Please do not hesitate to contact me if you have any questions or would like to discuss these revisions further.

Sincerely,

Giacomo Ariani
Editor
Nature Human Behaviour

Reviewer expertise:

Reviewer #1: Behavioral economics, Neuromodulators of decision making

Reviewer #2: Electrochemical recordings of dopamine or serotonin

Reviewer #3: Social cognition, Decision making

REVIEWER COMMENTS:

Reviewer #1:

Remarks to the Author:

I like this manuscript. It is a helpful and positive contribution to our understanding of monoaminergic influences on inter-individual social exchanges. The method is an innovative extension of electrochemistry into social neuroscience and the findings will be of interest to the broader cognitive neuroscience readership of NBH, pending significant revisions. Most of my concerns relate to the framing of the central social vs non-social comparison, understanding the significance of the results over and above what we know about DA/5-HT in other aspects of cognition, and aspects of the statistical analyses.

Major concerns.

1. I always struggle a bit with the manipulation of social cognition as comparisons between human partners in behavioural economic games versus computer partners as the control. This is because the latter offer pretty low baselines so that these designs tend to underspecify which aspects of social cognition and affect are likely to be most centrally involved.

In this particular experiment, the social condition cannot involve social learning since the protocol involves numerous presumably unique avatars presented with random and uncorrelated one-shot UG offers. By contrast, the latter non-social cognition involved a single – albeit non-human – playing partner that in principle could have offered participants the initial possibility of learning through engagement with a single agent over the games.

It is true that avatars must have invoked social processes through their visual humanoid presentation, to support the increased offer acceptance rates compared with the non-social/computer condition. However, this limits the interpretation of the signalling data. For example, the differences in DA activity shown in Fig. 3 could reflect the greater visual complexity of multiple avatars over the one-shot games compared with the single computer presentation (again not illustrated). In essence, the social context here seems to consist only in visually human-like partners playing independent games. The authors could improve the ms by better explaining why this manipulation is important, what it tells us about the role of DA/5-HT in social exchanges (and partnerships) beyond what we already know from other evidence and, drawing on the extant literature, the range of potential psychological mechanisms involved (e.g. calibrations of fairness, attributions, inhibition).

2. Notwithstanding the short NBH format, the descriptions of the statistical models and analyses are

too cursory. The models involve mixed effects logistic regressions of offer acceptances (vs rejections) against (i) condition (social/numerous avatars vs non-social/single computer); (ii) offer value and (iii) change in offer value (relative to the previous offer). However these models are not adequately described. Mixed effects models are complicated things and can be challenging both to specify and to test; the statistical choices are not straightforward. In this version of the ms, it is not clear exactly which predictors were specified as fixed and which as random (other than across 'dataset level'? Participant?). Tests are couched in terms of β -values and t-tests only; some with very high degrees of freedoms. There are no comparisons of models with and without the critical interactive terms, even against changes in R² (itself not the best way to determine between candidate models).

I also wondered whether about the covariances between offer value and change in offer value. It looks as if offer values over the one-shot games were randomly distributed between \$1 and \$9 (looking at Fig. 2) which presumably meant that the smaller offers tended to be followed by increased offers and the larger offers were followed by decreased offers. It would be helpful to see how these dependencies were handled and what the variances experienced by participants looked like and were handled by the models.

Finally, since the range of offers and range of change in offers would have been fixed across participants, it is not quite clear what was gained by the Z-score transformation for these predictors. Neither is it explained why the dependent measures of DA and 5-HT levels were transformed in the same way (other than by the loss of the intercept). All of the above is set out in mostly outline form that needs much better specification and justification, probably best in the supplementary materials to save on the main word count.

3. Interpreting the results, I was not quite convinced by the interpretation of the trial-by-trial changes in DA activity as being 'akin to a reward prediction errors'. I get why the authors have gone for this parallel as a link to the evidence on the role of DA signalling in reinforcement learning. But, as per the points above, these are one-shot games involving multiple partners in which offers were uncorrelated. Beyond a certain point, learning must have been minimal as participants came to understand that successive games and offers were independent. (One could argue that there was more potential for coding something like prediction errors in the computer condition that involved a single agent as a playing partner).

Since there is no computational model or behavioural evidence that participants are doing anything that might involve predictions, it might be better to adopt a more conservative and perhaps accurate characterisation of the DA and 5-HT as coding the variances in offers (and rewards) in different ways and scales; and then link their findings to older ideas about the broader and interdependent role of DA and 5-HT in the coding of reward and punishments in learning and choice behaviours. I appreciate the format allow only limited space for surveying large literatures like this but it should be possible to include some extra sentences to set these results with social processing in a wider context.

4. I wasn't convinced by the text sections (bottom of p.8) intended to allay concerns that these observations may partially reflect the pathophysiology of PD patients (p. 8, line 267 onward). First, we are not told much at all about the disease stage or any medications of these patents or what we know about the likely changes in DA and 5-HT function in patients who are judged suitable for DBS surgery. The dissociations between DA and 5-HT signalling in the social versus non-social settings does not change the fact that these observations could reflect broader pathological function. So, for example,

there are associations between essential tremor and Parkinson's Disease and the pathophysiology of the former certainly involves GABA and glutamate changes. Better arguments are needed or the revised text needs to be more conservative.

5. Finally, in places, the writing is a little careless and the manuscript read as if it had prepared in a rush with important or helpful details missing (e.g. aspects of the statistical modelling). It is true that the format has fairly restrictive word limit but the revision should be better prepared and us the figure legends and supplementary materials.

Minor points and tentative suggestions

Abstract.

p. 2, line 28. 'Participants rejected more offers in the human condition' compared with what?

p. 2, line 32. We don't know that DA and 5-HT are doing any 'broadcasting' (whatever that means) so 'reflects' might be better.

Introduction

The introductory sections could be improved by including a little more precision about the functional outputs between the SNc and SNr. Fig. 2a does not help much in this respect. If the authors wish to include anatomical diagram, it should tell us little but more about the inhibitory/excitatory aspects of the circuitry and innervation across the social brain.

p. 3, line 44. 'a similar breakthrough is yet to be made for the neuromodulatory systems that deliver chemical signals throughout the brain'

might be better as

'a similar breakthrough is yet to be made for the neuromodulatory systems that regulate activity across the networks of the social brain'

p. 3, line 49. 'One person, the "Proposer", splits up a monetary stake (e.g., \$20), and the other person, the "Responder", is to accept or reject'

could be edited as

'One person, the "Proposer", splits up a monetary stake (e.g., \$20) and makes an offer of all or only a portion of it for the "Responder" who can then accept or reject it.'

Typo/p. 4, line 90. 'offer' should 'offers'.

Why did the authors chose one-shot UGs rather than richer, more challenging iterated games that also engage monoamine systems (e.g. Woods et al, 2006)?

The main text needs to include an accessible summary of the modelling of SNr DA and 5-HT levels to connect it to the more detailed description in the Supplementary Materials.

Results

It isn't quite clear why DBS surgery requires two sessions? One for each hemisphere? Please clarify.

p. 4, lines 99-100. 'On around a third of trials, participants were asked to indicate how they felt about the game' is vague. It

could be better expressed as

'On around a third of trials, participants were asked to use a slider across a visual analog scale how positively or negatively they felt about the game'

p. 6, line 169-170. The sentence 'Overall estimates were computed as the sum across estimates within a 1-s window after offer presentation' is not quite clear. Figure legends do not enter into the word count so there is space for a fuller explanation.

I wondered whether it might be helpful if the outputs of the logistic models were specified as odds-ratios of accepted over rejected offers. (β s these models can be hard to interpret.)

p. 5, line 122. The presentation of the models is confusing. Model 1 includes the offer value, the condition (social vs non-social) and their interaction. Model 2 includes the difference between the current and previous offer. Don't we need some test of improved fit?

It might be helpful to explain why DA and 5-HT was tested with 1s windows. Is there a risk this means looking only at phasic rather than tonic changes?

Reviewer #2:

Remarks to the Author:

In this manuscript, the authors measure dopamine and serotonin in the human substantia nigra during an economic exchange task ('ultimatum game') in which players may accept or reject monetary offers of different fairness. The authors report that dopamine, but not serotonin, levels in subjects are higher during interaction with human players than with computer players, in line with social context. The authors also find that relative dopamine levels track with the changes in offer value, consistent with a 'reward prediction error' (RPE) role for dopamine. In contrast, relative serotonin levels scale with the content of the offer value itself (e.g. low offer value is associated with low serotonin). Together, these data suggest that dopamine and serotonin track different aspects of monetary offers and their value.

This is an interesting study that provides the first "real time" measurements of dopamine and serotonin during an economic choice task in humans. As such, it will be of interest to a variety of investigators studying the encoding of choice. The major caveats to this study are that measurements are performed in only a small number of Parkinsonian subjects, and in a brain area not often linked to RPE. However, the major findings on differences in dopamine/serotonin during these tasks are in line with a number of other preclinical and clinical studies, so the general validity of the results seems to apply.

My major question for the authors is whether or not it might be possible to provide some

supplementary details about the FSCV recordings. Specifically, there is no raw data anywhere in the manuscript, it is all transformed (z-score), making it challenging to evaluate the actual methods that were used to measure dopamine/serotonin. Because FSCV is inherently a background-subtracted technique, it's not clear how 'drift' of the background signal might have been accounted for. This may be less of an issue for relative changes (e.g. total DA/5-HT change in a 1s window relative to the offer period), but could be more difficult when looking at "overall" values (Methods, page 11, line 334). I would therefore suggest a supplemental figure that demonstrates a raw signal trace and how the signals were analyzed/summed/characterized. It sounds like a principal components approach may have been used, but again, this is not entirely clear in the Methods.

Reviewer #3:

Remarks to the Author:

In this work, the authors used a novel technique that allows them to track levels of serotonin and dopamine using an electrode inserted into the substantia nigra of awake DBS patients, and to track their activity during social ultimatum game. They found that dopamine levels were sensitive to social context of the game. They also found that dopamine levels followed the trial-by-trial changes in the financial offers, and that serotonin levels followed the absolute value of trial-by-trial financial offers. This study uses a novel technique in an elegant experimental design and provides novel and important findings.

It is well written, and statistical analyses are appropriate.

I have two comments:

1. Regarding the neuromodulator analyses:

- a. I am not sure I follow what happened to the social context effect in the dopamine and serotonin value and PE analysis – was it included in the regression as a control variable? Was it significant? Did you observe interactions between value/PE and condition?
- b. Did you observe any relation between response (accept/reject) and serotonin/dopamine? As participants were more likely to reject human offers, and dopamine was higher during human context, there could be a link between dopamine and acceptance responses.

2. Serotonin and behavior in ultimatum game – this study uses a technique with high temporal resolution and high spatial focus – the authors examine serotonin levels in SNr. As neuromodulators have large spread effect in different locations in the brain, it is possible that previous works that used pharmacological interventions found effects that were driven by serotonin in other parts of the brain, as their method was not as spatially focused. Is it possible that the effects of serotonin and dopamine in SNr may differ from their effect in other brain regions during social interaction? Maybe it is worth while to highlight the localized nature of the finding, and in general the idea that we should be more careful about stating roles of neuromodulators and neurotransmitters without the specific experimental context and specific brain region?

3. The finding about dopamine and social context is very interesting, and current works tie dopamine levels to social intentions judgement, especially in schizophrenia and paranoia (see recent works by Vaughan Bell and Nichola Raihani, Joe Barnaby, Michael Moutoussis, as well as Jennifer Cook and others). I think that this finding could be better highlighted in the context of these works, besides the RPE finding.

Author Rebuttal to Initial comments

Reviewer #1:

(1.1) I like this manuscript. It is a helpful and positive contribution to our understanding of monoaminergic influences on inter-individual social exchanges. The method is an innovative extension of electrochemistry into social neuroscience and the findings will be of interest to the broader cognitive neuroscience readership of NBH, pending significant revisions. Most of my concerns relate to the framing of the central social vs non-social comparison, understanding the significance of the results over and above what we know about DA/5-HT in other aspects of cognition, and aspects of the statistical analyses.

We thank the reviewer for the positive and constructive comments and hope that the changes described below address their concerns.

Major concerns.

(1.2) I always struggle a bit with the manipulation of social cognition as comparisons between human partners in behavioural economic games versus computer partners as the control. This is because the latter offers pretty low baselines so that these designs tend to underspecify which aspects of social cognition and affect are likely to be most centrally involved.

In this particular experiment, the social condition cannot involve social learning since the protocol involves numerous presumably unique avatars presented with random and uncorrelated one-shot UG offers. By contrast, the latter non-social cognition involved a single – albeit non-human – playing partner that in principle could have offered participants the initial possibility of learning through engagement with a single agent over the games.

It is true that avatars must have invoked social processes through their visual humanoid presentation, to support the increased offer acceptance rates compared with the non-social/computer condition. However, this limits the interpretation of the signalling data. For example, the differences in DA activity shown in Fig. 3 could reflect the greater visual complexity of multiple avatars over the one-shot games compared with the single computer presentation (again not illustrated). In essence, the social context here seems to consist only in visually human-like partners playing independent games. The authors could improve the ms by better explaining why this manipulation is important, what it tells us about the role of DA/5-HT in social exchanges (and partnerships) beyond what we already know from other evidence and, drawing on the extant literature, the range of potential psychological mechanisms involved (e.g. calibrations of fairness, attributions, inhibition).

We recognize that the reviewer is primarily encouraging us to consider these important points for the benefit of a general reader and have revised the manuscript accordingly.

We agree that it would be interesting to combine human electrochemistry with more complex social tasks, but we felt that the novelty of the method required the use of a task design which is simple and widely adopted. The one-shot version of the ultimatum game is perhaps the single most used social economic game, from anthropological investigations of cross-cultural variation in social fairness norms to foundational studies on the neural basis of human social behaviour. In addition, the human versus computer manipulation is a standard experimental approach for testing the specificity of neural signals in social neuroscience.

We understand the reviewer's reservations about the social manipulation, but we believe that low-level accounts are unlikely. First, in support of the human condition also engaging learning processes, previous studies have found that people form expectations about offer values even when presented with unique (human) avatars and randomized offers (e.g., Xiang et al., *Journal of Neuroscience*, 2013; Gu et al., *Journal of Neuroscience*, 2015). The fact that we here found a dissociation between brain and behavior – where dopamine, but not choice, was affected by the offer history – does not mean that no learning took place. Patients may have tracked offer values as indicated by the neural data but adopted fixed context-specific thresholds for offer acceptance. Second, in support of the human condition differently engaging social processes, previous studies have shown that the human condition is accompanied by increased affective arousal as measured by skin conductance (van 't Wout et al., *Experimental Brain Research*, 2006) and increased activity in the emotion-related brain regions, including amygdala, insula, and striatum (Sanfey et al., *Science*, 2006; Crockett et al., *Journal of Neuroscience*, 2023).

Finally, while our interpretation that dopamine helps set the stage for social interaction fits with earlier pharmacological results from economic games (e.g., Crockett et al., *Current Biology*, 2015; Pedroni et al., *Neuropsychopharmacology*, 2014; Sáez et al., *Current Biology*, 2015), we have not been able to find studies which have related slow changes in overall dopamine levels to low-level features such as visual complexity.

Changes:

(1.2.A) We now highlight that previous studies have found that people form expectations about offer values even when presented with unique avatars and randomized offers [lines 143-145]:

Previous studies have found that participants adapt their willingness to accept an offer to the history of offer values, including when offers are made by unique avatars and offer values are randomized^{36,37}. [...]

(1.2.B) We have expanded the discussion of the effect of social context on overall dopamine levels to provide a broader overview of the literature and consider a wider range of potential psychological mechanisms [lines 291-312]:

Our behavioral data replicated the result that people reject more offers when they believe they are interacting with another person as opposed to a computer²⁹⁻³². This effect of social context, which is accompanied by increased affective arousal as measured by skin conductance³¹ and increased activity in emotion-related brain regions (e.g., amygdala, insula, and striatum)^{29,32}, has been attributed to human social interaction invoking a sense of fairness. While rejecting “unfair” offers can promote cooperation⁴⁰, research suggests that our sense of fairness is in fact self-oriented: we view unfair offers as displays of dominance and reject them to avoid the imposition of inferior status⁴¹ or gain social control⁴². Such a change in the frame of reference for social interaction may explain why overall dopamine levels were higher for human than computer avatars. Indeed, pharmacological studies have found that elevated dopamine levels make people more averse to differences between their own and others’ payoffs⁴³, less averse to inflicting pain on others in exchange for money¹⁴, and more selfish when selfish behaviors cannot be punished⁴⁴. One prediction of the hypothesis that dopamine helps set the stage for social interaction is that disturbances in dopamine signaling should increase the risk of social dysfunction. Indeed, schizophrenia, associated with a dysregulated dopamine system⁴⁵, can involve delusions centered around social themes (e.g., persecutory delusions)^{46,47}, sometimes

ruining people's social lives. The hypothesis also fits with a growing literature linking dopamine to biases in social reasoning⁴⁸⁻⁵⁰, such as attribution of harmful intent. While pharmacological studies have found a link between overall serotonin levels and the willingness to accept unfair offers^{10,11}, dietary acute tryptophan depletion does not influence neural discrimination between human and computer conditions in the ultimatum game as assessed by fMRI²⁹. In line with this result, we found no effect of social context on overall serotonin levels.

(1.2.C) We have revised the final paragraph of the Discussion to explain the reasoning behind our choice of task and acknowledge the need for other social tasks [lines 363-375]:

Given the novelty of human electrochemistry, we decided to use a simple, widely used social task. For example, the one-shot version of the ultimatum game has been used to study cross-cultural variation in social fairness norms^{13,66}, the neural basis of social behavior³², and the social impact of brain injury^{36,67} and pharmacological manipulations of neuromodulators^{10,29} and hormones, such as oxytocin⁶⁸ and testosterone^{69,70}. Similarly, the human versus computer manipulation of social context has been used to assess social specificity beyond the ultimatum game^{29,32} in a range of social neuroscience studies^{71,72}. However, a complete understanding of the role of fast dopamine and serotonin signalling in human social behavior requires future experiments that involve repeated interaction and sophisticated inference, such as multi-round economic games⁷³⁻⁷⁵. In summary, our study provides the first direct evidence from the human brain that fast changes in dopamine and serotonin reflect context and value signals during social interaction, and that their distinct yet complementary roles in value coding generalize across contexts.

(1.3) Notwithstanding the short NBH format, the descriptions of the statistical models and analyses are too cursory. The models involve mixed effects logistic regressions of offer acceptances (vs rejections) against (i) condition (social/numerous avatars vs non-social/single computer); (ii) offer value and (iii) change in offer value (relative to the previous offer). However, these models are not adequately described. Mixed effects models are complicated things and can be challenging both to specify and to test; the statistical choices are not straightforward. In this version of the ms, it is not clear exactly which predictors were specified as fixed and which as random (other than across 'dataset level'? Participant?). Tests are couched in terms of β -values and t-tests only; some with very high degrees of freedoms. There are no comparisons of models with and

without the critical interactive terms, even against changes in R2 (itself not the best way to determine between candidate models).

We thank the reviewer for prompting us to clarify the statistical analysis including specification of mixed-effects models, unpack effects in simple terms and perform model comparisons when relevant for interpretation.

Changes:

(1.3.A) We now label the mixed-effects models for ease of reference. For example [line 130-133]:

To unpack participants' task behavior, we first ran a logistic mixed-effects model in which we predicted choices (reject = 0, accept = 1) using offer value, condition (computer = -.5, human = .5), and the interaction between these terms (choice model 1, C-M1; see **Methods** for details about statistical analysis of behavioral and neural data which was conducted at the trial level). [...]

(1.3.B) We now describe our statistical approach and specify all mixed-effects models in the “Statistical analysis” section in the Methods [line 422-448]:

Mixed-effects models

We used mixed-effects models specified at the trial level for statistical analysis of behavioral and neural data. All models included (1) fixed (population-level) effects and (2) random effects varying by dataset (session) with a free covariance matrix. We note that removing all random effects except for the intercept from a particular model did not change the significance of any of the reported effects for that model. We used the Akaike information criterion (AIC) for model comparison when relevant.

Behavioral models

In Wilkinson notation, the logistic choice (C; reject = 0, choice = 1) models were specified as:

C-M1, $C \sim 1 + \text{value} \times \text{condition} + (1 + \text{value} \times \text{condition} \mid \text{dataset})$

C-M2, $C \sim 1 + \text{condition} \times (\text{value} + \text{value difference}) + (1 + \text{condition} \times (\text{value} + \text{value difference}) \mid \text{dataset})$

The linear reaction time (RT) model was specified as:

RT-M1, $RT \sim 1 + \text{choice} \times \text{value} \times \text{condition} + (1 + \text{choice} \times \text{value} \times \text{condition} \mid \text{dataset})$

The emotion rating (E) model was specified as:

E-M1, $E \sim 1 + \text{choice} \times \text{value} \times \text{condition} + (1 + \text{choice} \times \text{value} \times \text{condition} \mid \text{dataset})$

Neural models

In Wilkinson notation, the linear neural (N) models were specified as:

N-M1, overall $N \sim 1 + \text{condition} \times \text{choice} + (1 + \text{condition} \times \text{choice} \mid \text{dataset})$

N-M2, overall $N \sim 1 + \text{condition} \times \text{choice} \times \text{order} + (1 + \text{condition} \times \text{choice} \times \text{order} \mid \text{dataset})$

N-M3, relative $N \sim 1 + \text{value} + \text{value difference} + (1 + \text{value} + \text{value difference} \mid \text{dataset})$

N-M4, relative $N \sim 1 + \text{value} + \text{value difference} + \text{absolute value difference} + (1 + \text{value} + \text{value difference} + \text{absolute value difference} \mid \text{dataset})$

N-M5, relative $N \sim 1 + \text{condition} \times (\text{value} + \text{value difference}) + (1 + \text{condition} \times (\text{value} + \text{value difference}) \mid \text{dataset})$

For N-M3*, which controls for correlations between value and value difference, we regressed value difference against value and used the residuals as our predictor for value difference.

(1.3.C) We now unpack reported effects in simple terms. For example [line 134-141]:

[...] As expected, value had a positive effect on choice (positive slope in **Fig. 2a**; C-M1, value, $\beta \pm \text{SE} = 1.77 \pm 0.73$, $t(454) = 2.43$, $p = .016$): participants accepted 43% of offers smaller than, or equal to, one standard deviation (\$2) below the mean (\$5), 58% of offers within one standard deviation of the mean, and 94% of offers equal to, or larger than, one standard deviation above the mean. In line with earlier results²⁹⁻³², condition had a negative effect on choice (pink below cyan in **Fig. 2a**; C-M1, condition, $\beta \pm \text{SE} = -3.60 \pm 1.61$, $t(454) = -2.23$, $p = .026$): participants accepted 50% of the offers made by human avatars but 75% of those made by the computer avatar. [...]

(1.3.D) We now perform model comparisons using the Akaike information criterion (AIC) when relevant for interpretation.

When evaluating the choice model with history effects, we now say [line 151-153]:

[...] Nevertheless, indicative of some behavioral adaptation being at play, the model which included history effects provided a better fit to the data than the basic model (AIC, C-M1 = 3016, C-M2 = 2844). [...]

When evaluating whether order effects may have confounded the effect of social context on overall levels of dopamine, we now say [line 214-216]:

[...] Consistent with an absence of order-related effects, the model which included order effects provided a worse fit to the data compared to the basic model (AIC; dopamine, N-M1 = 1205, N-M2 = 1262; serotonin, N-M1 = 1269, N-M2 = 1331). [...]

When evaluating whether relative changes in dopamine/serotonin reflect context-independent value signals, we now say [line 274-277]:

[...] In line with an absence of modulation by social context, the model itself provided a worse fit to the data compared to the basic model (AIC; dopamine, N-M3 = 1230, N-M5 = 1262; serotonin, N-M3 = 1231, N-M5 = 1273).

(1.4) I also wondered whether about the covariances between offer value and change in offer value. It looks as if offer values over the one-shot games were randomly distributed between \$1 and \$9 (looking at Fig. 2) which presumably meant that the smaller offers tended to be followed by increased offers and the larger offers were followed by decreased offers. It would be helpful to see how these dependencies were handled and what the variances experienced by participants looked like and were handled by the models.

The reviewer is right that the randomization of offer values within a bounded range means that smaller offers tend to be followed by larger offers and vice versa. In our data, value and value difference are indeed correlated (Pearson's $r = .73$; note that we computed value difference as the difference between the current and the previous trial, and not as the difference between the next and the current trial, which would have returned a negative correlation). However, the logistic mixed-effects model including history effects identified an effect of value but not value difference, indicating that the correlation between the variables is not an issue for model fitting. We note that, even if we re-run this model without value, then the analysis does not return an effect of value difference. In other words, while value and value difference are correlated, there is a substantial number of trials that are ranked differently along the two metrics. Nevertheless, because the link between these variables and relative changes in dopamine/serotonin is one of the key results, we have now conducted an analysis that controls for the correlation between value and value difference. In particular, we regressed value difference against value and used the residuals (Pearson's $r \sim 0$) as our predictor for value difference (N-M3*). Critically, this analysis returned an effect of value difference on dopamine and an effect of value on serotonin.

Changes:

(1.4.A) We now report and control for the correlation between value and value difference when analyzing relative dopamine/serotonin [line XXX-XXX]:

[...] To control for any model misestimation due to the correlation between value and value difference (Pearson's $r = .73$), we regressed value difference against value and used the residuals (Pearson's $r \sim 0$) as our predictor for value difference (N-M3*). In keeping with the original results, this analysis returned an effect of value difference on dopamine (N-M3*; value, $\beta \pm SE = 0.05 \pm 0.06$, $t(426) = -1.47$, $p = .412$; value difference, $\beta \pm SE = 0.21 \pm 0.07$, $t(426) = 2.82$, $p = .005$) and an effect of value on serotonin (N-M3*; value, $\beta \pm SE = 0.13 \pm 0.05$, $t(426) = 2.67$, $p = .008$; value difference, $\beta \pm SE = -0.13 \pm 0.07$, $t(426) = -1.82$, $p = .070$).

(1.5) Finally, since the range of offers and range of change in offers would have been fixed across participants, it is not quite clear what was gained by the Z-score transformation for these predictors. Neither is it explained why the dependent measures of DA and 5-HT levels were transformed in the same way (other than by the loss of the

intercept). All of the above is set out in mostly outline form that needs much better specification and justification, probably best in the supplementary materials to save on the main word count.

We apologize for the lack of clarity and thank the reviewer for prompting us to better describe how and why we standardized the behavioral and neural data.

Changes:

(1.5.A) We now explain the coding and standardization of variables in the “Statistical analysis” section in the Methods [line 449-462]:

Coding and standardization of variables

Binary variables were contrast coded (-.5 or .5) and continuous variables were standardized separately for each dataset using a z-score transformation. We performed standardization, which transforms data into a relative frame of reference, for several reasons. First, it facilitates the comparison of fitted coefficients within a given model. For example, while value difference is derived from value, their fitted coefficients cannot be compared without standardization as the raw variables have different means and variances. Second, it facilitates the comparison of fitted coefficients when the same model is applied to different data. For example, if value had been found to have a positive effect on both relative dopamine and serotonin, then the fitted coefficients could be compared across the neuromodulators. Third, in the case of neural data, standardization minimizes, if not removes, the influence of any unmodelled sources of dataset-level variation in the baseline and/or the variance of the data. Finally, standardization mitigates against between-dataset differences due to trial exclusions, which again can affect the mean and/or the variance of the data.

(1.6) Interpreting the results, I was not quite convinced by the interpretation of the trial-by-trial changes in DA activity as being 'akin to a reward prediction errors'. I get why the authors have gone for this parallel as a link to the evidence on the role of DA signalling in reinforcement learning. But, as per the points above, these are one-shot

games involving multiple partners in which offers were uncorrelated. Beyond a certain point, learning must have been minimal as participants came to understand that successive games and offers were independent. (One could argue that there was more potential for coding something like prediction errors in the computer condition that involved a single agent as a playing partner).

Since there is no computational model or behavioural evidence that participants are doing anything that might involve predictions, it might be better to adopt a more conservative and perhaps accurate characterisation of the DA and 5-HT as coding the variances in offers (and rewards) in different ways and scales; and then link their findings to older ideas about the broader and interdependent role of DA and 5-HT in the coding of reward and punishments in learning and choice behaviours. I appreciate the format allow only limited space for surveying large literatures like this but it should be possible to include some extra sentences to set these results with social processing in a wider context.

We believe that it is reasonable to consider learning-related effects since – as described in **response 1.2** – earlier studies have found that people form expectations about offer values even when presented with unique avatars and randomized offers (e.g., Xiang et al., *Journal of Neuroscience*, 2013; Gu et al., *Journal of Neuroscience*, 2015). As we also explain, the fact that we here found a dissociation between brain and behavior – where dopamine, but not choice, was affected by the offer history – does not mean that no learning took place. Patients may have tracked offer values as indicated by the neural data but adopted fixed context-specific thresholds for offer acceptance. Furthermore, we believe that that it is appropriate to interpret the dopamine results in terms of RPE-signalling given the substantial literature relating dopamine to RPEs as conceptualized by RL and our definition of value difference being a one-step RPE with a learning rate = 1. That being said, following the reviewer’s suggestion, we have run an analysis that probes coding of RPEs versus variability. Inspired by the recent hypothesis that serotonin tracks absolute (unsigned) RPEs, which reflect variability in the environment and can be used to regulate learning (Matias et al., *eLife*, 2017; Grossman et al., *Current Biology*, 2023), we included the absolute value difference as an additional predictor. Indicating that our results do not reflect variability, the analysis replicated the original results but did not return an effect of absolute value difference for dopamine or serotonin (see **change 1.6.A**).

Changes

(1.6.A) We have run an analysis that probes coding of RPEs versus variability [line 253-265]:

Recent research suggests that serotonin tracks absolute (unsigned) RPEs, which provide an estimate of variability in the environment and can be used to regulate the rate of learning^{19,38}. Since the visualization of relative changes in serotonin was consistent with this computational function (U-shape in **Fig. 4**, bottom right), we tested this hypothesis formally, by also including the absolute value difference as a predictor (NM-4). However, while this analysis replicated the original results, it did not identify an effect of absolute value difference for dopamine (NM-4; value, $\beta \pm SE = -0.11 \pm 0.07$, $t(425) = -0.77$, $p = .444$; value difference, $\beta \pm SE = 0.21 \pm 0.07$, $t(425) = 2.87$, $p = .004$; absolute value difference, $\beta \pm SE = 0.08 \pm 0.09$, $t(425) = 0.92$, $p = .360$) or serotonin (N-M4; value, $\beta \pm SE = 0.22 \pm 0.07$, $t(425) = 3.07$, $p = .002$; value difference, $\beta \pm SE = -0.13 \pm 0.07$, $t(425) = -1.78$, $p = .076$; absolute value difference, $\beta \pm SE = 0.02 \pm 0.09$, $t(426) = 0.18$, $p = .861$). In addition, the model itself provided a worse fit to the data compared to the basic model AIC; dopamine, N-M3 = 1230, N-M4 = 1242; serotonin, N-M3 = 1231, N-M4 = 1244).

(1.6.B) We now explicitly consider the brain-behavior dissociation when discussing the value-related results [line 325-328]:

[...] We highlight that the brain-behavior dissociation – where dopamine, but not choice, was affected by task history – does not mean that no learning took place. Patients may have tracked offer values as indicated by the neural data but adopted fixed context-specific thresholds for offer acceptance.

(1.7) I wasn't convinced by the text sections (bottom of p.8) intended to allay concerns that these observations may partially reflect the pathophysiology of PD patients (p. 8, line 267 onward). First, we are not told much at all about the disease stage or any medications of these patients or what we know about the likely changes in DA and 5-HT function in patients who are judged suitable for DBS surgery. The dissociations between DA and 5-HT signalling in the social versus non-social settings does not change the fact that these observations could reflect broader pathological function. So, for example, there are associations between essential tremor and Parkinson's Disease and the

pathophysiology of the former certainly involves GABA and glutamate changes. Better arguments are needed or the revised text needs to be more conservative.

We thank the reviewer for prompting us to provide more clinical information about the patients who participated in our study, including motor symptoms, cognitive function and medications. Importantly, the patients' disease progression was not so severe that DBS would have been considered unlikely to be effective, the patients had a normal range of cognitive function, and the results were found despite patients receiving a diverse range of medications. We believe that this information strengthens our case that the results reflect general functions of dopamine and serotonin, and not the pathophysiology of Parkinson's disease patients, which are, of course, by no means an homogenous group.

(1.7.A) We have created **Supplementary Table 1** which provides patient information:

Patients		1	2	3	4	
Demographic information	Age	80	65	72	69	
	Sex	F	M	M	M	
	Race	White	White	White	White	
	Primary Diagnosis	PD/postural instability	PD	PD/tremor dominant	PD/tremor dominant	
PD severity	MDS-UPDRS-III	On	6 (Mild)	20 (Mild)	20 (Mild)	15 (Mild)
		Off	30 (Mild)	24 (Mild)	55 (Moderate)	32 (Mild)
Psychiatric symptoms	BDI-II	10 (Normal)	10 (Normal)	3 (Normal)	7 (Normal)	
	BAI	23 (Moderate)	10 (Normal)	10 (Normal)	10 (Normal)	
	Starkstein Apathy Scale	7 (Normal)	9 (Normal)	3 (Normal)	10 (Normal)	

Cognitive ability	WASI-II Similarities	24 (Average)	24 (Low average)	21 (Low average)	34 (High average)
	TOPF	65 (Superior)	59 (High average)	Test not administered	58 (High average)
Medication history	Carbidopa (mg)	325	112.5	300	350
	Levodopa (mg)	1300	450	1200	1400
	Other Medications	lorazepam sertraline gabapentin mirtazapine docusate sodium	amantadine quetiapine losartan pravastatin	pramipexole entecavir rosuvastatin tacrolimus	entacapone amantadine mirtazapine fludrocortisone infiximab baclofen

Supplementary Table 1. Patient information. Patients had mild to moderate Parkinson’s disease (PD) as measured using MDS-UPDRS-III¹. Patients did not meet diagnostic thresholds for depression, anxiety, or apathy as measured using BDI-II², BAI³, and the Starkstein Apathy Scale⁴, respectively. Patients had a range of cognitive scores from low average to superior as measured using WASI-II Similarities⁵ and TOPF⁶. All patients received dopamine replacement therapy (carbidopa and levodopa) but different Parkinson and psychiatric medications; all patients were off dopamine replacement therapy and Parkinson medications during the surgical sessions.

(1.7.B) We have expanded the Discussion to include points concerning disease progression, cognitive function, and medications [line 346-362]:

Our electrochemical data necessarily had to be collected in brain surgery patients; in our case, Parkinson’s disease patients undergoing bilateral DBS surgery. While Parkinson’s disease is characterized by a loss of midbrain dopamine neurons⁶³, there are several reasons why our results are likely to generalize to the healthy brain. First, the patients’ disease progression was not so severe that DBS would have been unlikely to be effective (Supplementary Table S1). Second, even though the patients may have a general reduction in dopamine levels, then it should not be able to explain the difference between conditions in our within-subject design. Third, indicative of an otherwise normal range of brain function, the patients did not present with significant cognitive impairment or refractory psychiatric disorders (Supplementary Table S1), both contraindications for DBS. Fourth, our results are unlikely to be confounded by medication considering that Parkinson medication was withheld during surgery and that the patients otherwise received different types of medications (Supplementary Table S1). Fifth, previous studies applying human electrochemistry during DBS surgery have seen comparable dopamine and serotonin responses in Parkinson’s disease and essential tremor²², with the latter condition

involving small, or no, disturbances in the dopamine and serotonin systems⁶⁴. Finally, the value-related results for dopamine are consistent with a large body of animal work on RPE signaling in the basal ganglia⁶⁵.

(1.8) Finally, in places, the writing is a little careless and the manuscript read as if it had prepared in a rush with important or helpful details missing (e.g. aspects of the statistical modelling). It is true that the format has fairly restrictive word limit but the revision should be better prepared and us the figure legends and supplementary materials.

We apologize for any inaccuracies. We hope that the changes to the manuscript provide much greater clarity about the methods, including the statistical analyses, and hope that the changes address the reviewer's concerns.

Minor points and tentative suggestions

Abstract.

(1.9) p. 2, line 28. 'Participants rejected more offers in the human condition' compared with what?

Changes:

(1.9.A) We have clarified the text [line 28-29]:

[...] They rejected more offers in the human than the computer condition, [...]

(1.10) p. 2, line 32. We don't know that DA and 5-HT are doing any 'broadcasting' (whatever that means) so 'reflects' might be better.

Changes:

(1.10.A) We have revised the text as suggested [line 32-33]:

[...] These results show that dopamine and serotonin fluctuations in one of the basal ganglia's main output structures reflect distinct social context and value signals.

Introduction

(1.11) The introductory sections could be improved by including a little more precision about the functional outputs between the SNc and SNr. Fig. 2a does not help much in this respect. If the authors wish to include anatomical diagram, it should tell us little but more about the inhibitory/excitatory aspects of the circuitry and innervation across the social brain.

Changes:

(1.11.A) We have replaced the "mid-brain" cartoon in **Fig. 1** with a "whole-brain" cartoon which illustrates the electrode trajectory and the recording site:

Fig. 1. Experimental framework. **a** Illustration of electrode trajectory and recording site. GP: globus pallidus. STN: sub-thalamic nucleus. SNr: substantia nigra pars reticulata. SNc: substantia nigra pars compacta. RN: raphe nucleus. Brain slice created with <http://Biorender.com>. **b** The game involved 60 trials of one-shot ultimatum games where participants had to accept or reject splits of a \$20 stake proposed by either a human (30 trials) or a computer (30 trials) avatar. On around a third of trials, participants were asked to indicate how they felt about the game. The human and computer conditions were blocked within a surgical session and their order was counterbalanced across surgical sessions.

(1.11.B) We have added a paragraph to the Discussion which reviews the connections of the SNr and considers the regional specificity of the results [line 329-345]:

Our electrochemical data were collected from the SNr (**Fig. 1a**); we should therefore consider (1) its anatomical connections and (2) whether our results are specific to the SNr or reflect signals that are broadcasted widely within the brain. First, the SNr is one of the basal ganglia's main output nuclei: it receives excitatory glutamatergic inputs from the subthalamic nucleus⁵⁴, inhibitory GABAergic inputs from the striatum⁵⁴, dopaminergic inputs from substantia nigra pars compacta (SNc)⁵⁵, and serotonergic inputs from the raphe nucleus^{56,57}; and it sends GABAergic outputs to the thalamus⁵⁴ which control glutamatergic outputs from the thalamus – a main relay station for sensorimotor information – to cortical and sub-cortical regions^{54,58–61}. These distal projection targets include regions that support decision-making in non-social and social contexts, including orbitofrontal cortex, medial prefrontal cortex, anterior cingulate cortex, and the amygdala. Second, it is hard to say whether our results are specific to the SNr. Dopamine

release in the SNr is mainly driven by somatodendritic release from the SNc⁵⁵, but this mechanism can be activated by action potentials in the SNc which drive synaptic release in other brain regions⁶². Similarly, while serotonin release in the SNr is mainly driven by direct synaptic release from the raphe nucleus^{56,57}, the upstream serotonergic neurons may project to other brain regions. Future research using human electrochemistry could offer insight about regional specificity by recording from multiple brain regions on the same task.

(1.12) p. 3, line 44. 'a similar breakthrough is yet to be made for the neuromodulatory systems that deliver chemical signals throughout the brain'

might be better as

'a similar breakthrough is yet to be made for the neuromodulatory systems that regulate activity across the networks of the social brain'

Changes:

(1.12.A) We have revised the text essentially as suggested [line 44-45]:

[...] but a similar breakthrough has yet to be made for the neuromodulatory systems that regulate activity across the social brain.

(1.13) p. 3, line 49. 'One person, the “Proposer”, splits – up a monetary stake (e.g., \$20), and the other person, the “Responder”, is to accept or reject'

could be edited as

'One person, the "Proposer", splits up a monetary stake (e.g., \$20) and makes an offer of all or only a portion of it for the "Responder" who can then accept or reject it.'

Changes:

(1.13.A) We have revised the text to further clarify the task [line 49-51]:

[...] A "Proposer" offers a split of a monetary stake (e.g., \$20) to a "Responder" who can then accept or reject the split. The Proposer can make any offer, from keeping to sharing the full stake. [...]

(1.14) Typo/p. 4, line 90. 'offer' should 'offers'.

Changes:

(1.14.A) We have corrected the typo [line 89-90]:

[...] participants rejected more human than computer offers. [...]

(1.15) Why did the authors chose one-shot UGs rather than richer, more challenging iterated games that also engage monoamine systems (e.g. Woods et al, 2006)?

See **response 1.2**.

(1.16) The main text needs to include an accessible summary of the modelling of SNr DA and 5-HT levels to connect it to the more detailed description in the Supplementary Materials.

Changes:

(1.16.A) We have added a short summary of our electrochemical approach when introducing our experimental framework [line 107-115]:

[...] Our electrochemistry protocol, which builds on earlier work in both animals^{33,34} and humans^{22-25,35}, provides 10 samples per second. In brief, the protocol involves the repeated delivery of a rapid change in electrical potential to a carbon-fiber electrode and measurement of induced electrochemical reactions as changes in current at the electrode tip. The current responses carry information not only about the identity but also the concentration of neuromodulators in the surrounding neural tissue. This information is extracted using a signal prediction model trained on large wet-lab datasets where the chemical environment can be carefully controlled (see **Supplementary Fig. 1** for an illustration of the electrochemical approach as well as in-vitro evaluation of signal prediction model; see **Methods** for details).

Results

(1.17) It isn't quite clear why DBS surgery requires two sessions? One for each hemisphere? Please clarify.

Changes:

(1.17.A) We have revised the text to clarify why there were two surgical sessions [116-118]:

Each participant performed the ultimatum game in two sessions; the clinical treatment involved two separate surgeries 14-28 days apart for the bilateral implantation of DBS electrodes in the subthalamic nucleus of each hemisphere (4 patients x 2 sessions = 8 datasets). [...]

(1.18) p. 4, lines 99-100. 'On around a third of trials, participants were asked to indicate how they felt about the game' is vague. It

could be better expressed as

'On around a third of trials, participants were asked to use a slider across a visual analog scale how positively or negatively they felt about the game'

Changes:

(1.18.A) We have revised the text [line 126-128]:

[...] On around a third of trials, participants were asked to rate how they felt about the game, by moving a slider along a visual analog mood scale ranging from negative (sad emoji) to positive (happy emoji).

(1.19) p. 6, line 169-170. The sentence 'Overall estimates were computed as the sum across estimates within a 1-s window after offer presentation' is not quite clear. Figure legends do not enter into the word count so there is space for a fuller explanation.

Changes:

(1.19.A) We have revised the text for **Fig. 3** which – together with **change 1.16.A** – should now clarify how the overall neuromodulator estimates were calculated [line 188-189]:

[...] Overall estimates were computed as the sum of neuromodulator samples within a 1-s window (10 samples) after offer presentation. [...]

(1.20) I wondered whether it might be helpful if the outputs of the logistic models were specified as odds-ratios of accepted over rejected offers. (βs these models can be hard to interpret.)

As detailed in **change 1.3.C**, we now explain the effects in simple behavioral terms.

(1.21) p. 5, line 122. The presentation of the models is confusing. Model 1 includes the offer value, the condition (social vs non-social) and their interaction. Model 2 includes the difference between the current and previous offer. Don't we need some test of improved fit?

As detailed in **change 1.3.D**, we now run model comparisons when relevant for interpretation – in this case finding that the logistic mixed-effects model with history effects provides a better fit to the data.

(1.22) It might be helpful to explain why DA and 5-HT was tested with 1s windows. Is there a risk this means looking only at phasic rather than tonic changes?

Changes:

(1.22.A) We now describe in **Fig. 3** why we used a 1-s window for the analysis of overall levels of dopamine/serotonin [line 189-192]:

[...] We limited the estimates to this window for consistency with the relative analysis in **Fig. 4** and to ensure that all estimates were based on the same number of samples regardless of variation in reaction times and trial events (e.g., variable duration of proposer screen and emotion ratings). [...]

(1.22.B) We now also report in **Fig. 3** that the results for overall levels of dopamine/serotonin are not limited to a particular time window [line 192-194]:

[...] We highlight, however, that the effect of condition on dopamine remained regardless of the specific time window (e.g., a 6-s window centered on offer presentation, $\beta \pm SE = 1.00 \pm 0.38$, $t(454) = 2.63$, $p = .009$). [...]

Reviewer #2:

(2.1) In this manuscript, the authors measure dopamine and serotonin in the human substantia nigra during an economic exchange task ('ultimatum game') in which players

may accept or reject monetary offers of different fairness. The authors report that dopamine, but not serotonin, levels in subjects are higher during interaction with human players than with computer players, in line with social context. The authors also find that relative dopamine levels track with the changes in offer value, consistent with a ‘reward prediction error’ (RPE) role for dopamine. In contrast, relative serotonin levels scale with the content of the offer value itself (e.g. low offer value is associated with low serotonin). Together, these data suggest that dopamine and serotonin track different aspects of monetary offers and their value.

This is an interesting study that provides the first “real time” measurements of dopamine and serotonin during an economic choice task in humans. As such, it will be of interest to a variety of investigators studying the encoding of choice. The major caveats to this study are that measurements are performed in only a small number of Parkinsonian subjects, and in a brain area not often linked to RPE. However, the major findings on differences in dopamine/serotonin during these tasks are in line with a number of other preclinical and clinical studies, so the general validity of the results seems to apply.

We thank the reviewer for the positive assessment and hope that the changes described below provide greater clarity about our electrochemical approach.

(2.2) My major question for the authors is whether or not it might be possible to provide some supplementary details about the FSCV recordings. Specifically, there is no raw data anywhere in the manuscript, it is all transformed (z-score), making it challenging to evaluate the actual methods that were used to measure dopamine/serotonin. Because FSCV is inherently a background-subtracted technique, it’s not clear how ‘drift’ of the background signal might have been accounted for. This may be less of an issue for relative changes (e.g. total DA/5-HT change in a 1s window relative to the offer period), but could be more difficult when looking at “overall” values (Methods, page 11, line 334). I would therefore suggest a supplemental figure that demonstrates a raw signal trace and how the signals were analyzed/summed/characterized. It sounds like a principal components approach may have been used, but again, this is not entirely clear in the Methods.

We apologize for the lack of clarity and thank the reviewer for prompting us to describe the electrochemical approach in a general manner. As described in **change 2.2.B**, the approach builds on FSCV as applied in model organisms but involves several innovations – which have been validated in previous work. The in-vivo data analysis is based on signal predictions (e.g., DA = 450 nM and 5-HT = 650 nM) from a model trained on large in-vitro datasets (7,260 unique concentration combinations and 1,089,000 current sweeps). The predictions are based on the full (differentiated) current response; we do not apply background subtraction or decompose the current response into principal components.

Changes:

(2.2.A) We have added a short summary of our electrochemical approach when introducing our experimental framework [line 107-115]:

[...] Our electrochemistry protocol, which builds on earlier work in both animals^{33,34} and humans^{22–25,35}, provides 10 samples per second. In brief, the protocol involves the repeated delivery of a rapid change in electrical potential to a carbon-fiber electrode and measurement of induced electrochemical reactions as changes in current at the electrode tip. The current responses carry information not only about the identity but also the concentration of neuromodulators in the surrounding neural tissue. This information is extracted using a signal prediction model trained on large wet-lab datasets where the chemical environment can be carefully controlled (see **Supplementary Fig. 1** for an illustration of the electrochemical approach as well as in-vitro evaluation of signal prediction model; see **Methods** for details).

(2.2.B) We have added a general description of the electrochemical approach to the Methods [line 474-519]:

General description

Human electrochemistry^{22–25} builds on fast-scan cyclic voltammetry (FSCV) as used in animal work^{76,77}. The carbon-fiber electrodes are made in the same way as those used in rodents⁷⁸, except with dimensions modified for use in the human brain⁷⁹. The data acquisition protocol is similar to that used in rodents with regards to the time course of the voltage sweeps and the recording of the induced current time series during those sweeps⁸⁰. The main change from animal work is the statistical method used to estimate the concentration of analytes of interest from the measured current time series (**Supplementary Fig. 1**).

In brief, FSCV involves the delivery of a rapid change in electrical potential to an electrode and measurement of the induced electrochemical reactions as changes in current at the electrode tip – with the guiding idea being that the current response carries information about both the identity and the concentration of analytes in the surrounding neural tissue. The goal of an analysis method for FSCV data is therefore to develop a statistical model that uses the current response in the best possible way to separate and estimate analytes of interest. The standard procedure is to train the statistical model on in-vitro data collected in the laboratory where the presence and concentration of analytes of interest can be controlled and then apply this model to in-vivo data for signal prediction.

Traditionally, the statistical model involves a decomposition of the in-vitro training data into principal components that are then used for in-vivo analyte inference within a regression framework⁸¹. In broad terms, this approach treats analyte inference as a problem of signal reconstruction: the concentration of an analyte of interest is estimated by mapping an in-vivo current response onto those collected in-vitro and then using the best match to label the in-vivo current response. We instead treat analyte inference as a problem of signal prediction, with the statistical model optimized to generate accurate predictions about out-of-training data. Previous human work^{22,24,25} has used elastic net regression⁸², but recent years have seen the development of more powerful machine learning methods. Here, we used deep convolutional neural networks as described below. Since information is distributed throughout a current time series, and not only at the oxidation or reduction peaks revealed by principal components analysis^{22,24,25}, we use non-decomposed data such that every time point within a current time series contributes to signal prediction. To facilitate out-of-training prediction, we train the model using large in-vitro datasets and cross-validation as described below.

There are statistical advantages to this approach to analyte inference. First, cross-validated training mitigates against any bias in the assembly of the training data and prevents against overfitting to the training data. Second, reframing analyte inference as a problem of signal prediction means that the statistical model can be directly evaluated using in-vitro data that were withheld from training. Third, an objective classification approach sidesteps the need for experimenter judgement (e.g., the cut-off for the number of principal components based on their reconstructed variance) and visual assessment of current responses (e.g., visualisation of background-subtracted voltammograms).

Earlier work has taken steps to validate human electrochemistry. First, the human-compatible carbon-fiber electrodes have similar electrochemical properties to those used in the rodents⁷⁹. Second, the signal prediction approach returns more reliable neuromodulator estimates than principal component regression²⁴. Third, it does not confuse changes in pH for changes in neuromodulators^{22,24,25}. Fourth, it does not confuse neuromodulators with one another^{22,24,25,35}. Fifth, it returns accurate neuromodulator estimates when tested in a laboratory setting where two neuromodulators simultaneously change across time²².

(2.2.C) We have revised **Supplementary Fig. 1** to provide more details and a better intuition for the electrochemical approach:

Supplementary Fig. 1. Electrochemical approach. **a** The data acquisition protocol is based on FSCV. We applied a triangular voltage waveform at 10 Hz. **b** We measured current during the application of the triangular voltage waveform. The current response carries information about the identity and the concentration of analytes in the surrounding neural tissue. **c** We did not apply background subtraction or decompose the current response into principal components; instead, we used the full differentiated current response for signal prediction. **d** The signal prediction model was trained and tested on in-vitro datasets from 64 carbon-fiber electrodes; 59 datasets were used for training and 5 datasets were used for evaluation. For each dataset, dopamine (DA), serotonin (5-HT), norepinephrine (NE), and pH were varied in small increments and multiple measurements were made at each step. In total, the training set consisted of 7,260 unique concentration combinations and 1,089,000 current sweeps, and the test set consisted of 795 unique concentration combinations and 119,250 current sweeps. **e** The signal prediction model is based on convolutional neural networks. **f** The signal prediction model generates concurrent predictions about DA, 5-HT, NE, and pH for each current sweep. **g** In-vitro evaluation of the signal prediction model was performed using the datasets withheld from model training. The evaluation shows predicted concentration (y-axis) as a function of labelled concentration (x-axis) for DA, 5-HT, and NE. Black line indicates the “x = y” identity line. Mean data are shown.

(2.2.D) We note that, as described in **change 1.22.B**, the results for overall levels of dopamine/serotonin are not limited to a particular time window [line 192-194]:

[...] We highlight, however, that the effect of condition on dopamine remained regardless of the specific time window (e.g., a 6-s window centered on offer presentation, $\beta \pm SE = 1.00 \pm 0.38$, $t(454) = 2.63$, $p = .009$). [...]

Reviewer #3:

(3.1) In this work, the authors used a novel technique that allows them to track levels of serotonin and dopamine using an electrode inserted into the substantia nigra of awake DBS patients, and to track their activity during social ultimatum game. They found that dopamine levels were sensitive to social context of the game. They also found that dopamine levels followed the trial-by-trial changes in the financial offers, and that serotonin levels followed the absolute value of trial-by-trial financial offers.

This study uses a novel technique in an elegant experimental design and provides novel and important findings.

It is well written, and statistical analyses are appropriate.

We thank the reviewer for the positive comments and the helpful suggestions for how to better contextualize our results.

I have two comments:

(3.2) Regarding the neuromodulator analyses:

- a. I am not sure I follow what happened to the social context effect in the dopamine and serotonin value and PE analysis – was it included in the regression as a control variable? Was it significant? Did you observe interactions between value/PE and condition?
- b. Did you observe any relation between response (accept/reject) and serotonin/dopamine? As participants were more likely to reject human offers, and dopamine was higher during human context, there could be a link between dopamine and acceptance responses.

We apologize for the lack of clarity and thank the reviewer for prompting us to revisit the presentation of the results.

For (a), we did include condition and its interactions with value and value difference when analyzing changes in dopamine/serotonin relative to a local baseline (**Fig. 4**), but we did not observe any main or interaction effect relating to condition. These results fit with a hypothesis that relative dopamine/serotonin reflect generalized value signals.

For (b), we did include choice and its interaction with condition when analyzing overall levels of dopamine/serotonin (**Fig. 3**), but we did not observe any main or interaction effects relating to choice. In the case of dopamine, the presence of a block-level effect of condition but the absence of trial-level effects of choice suggest that overall dopamine levels set the stage for social interaction – potentially driving a general change in the willingness to accept an offer made by a human versus a computer (**Fig. 2a**) – but does not drive individual choices per se.

Changes:

(3.2.A) As detailed in **changes (1.3.A-B)**, we now label the mixed-effects models and specify them in the Methods, which should clarify which predictors were included in each analysis.

(3.2.B) We now provide more context for the analysis where we ask whether the value-related signals carried by relative dopamine/serotonin are modulated by condition [line 266-277]:

Finally, we asked whether the value-related effects were modulated by social context. To this end, we re-ran the linear mixed-effects model after including condition and its interaction with the value-related terms as predictors (N-M5). In support of a hypothesis that relative changes in dopamine and serotonin reflect generalized value signals, the analysis replicated the earlier effects, but did not identify any condition-related effects, for dopamine (N-M5; value, $\beta \pm SE = -0.11 \pm 0.08$, $t(423) = -1.36$, $p = .176$; value difference, $\beta \pm SE = 0.21 \pm 0.08$, $t(423) = 2.63$, $p = .009$; condition-related effects, all absolute $t(423) < 0.93$, all $p > .354$) or serotonin (N-M5; value, $\beta \pm SE = 0.21 \pm 0.07$, $t(423) = 3.00$, $p = .003$; value difference, $\beta \pm SE = -0.12 \pm 0.07$, $t(423) = -1.72$, $p = .086$; condition-related effects, all absolute $t(423) < 0.86$, all $p > .390$). In line with an absence of modulation by social context, the model itself provided a worse fit to the data compared to the basic model (AIC; dopamine, N-M3 = 1230, N-M5 = 1262; serotonin, N-M3 = 1231, N-M5 = 1273).

(3.2.C) We now provide more context for the analysis where we assess the effects of condition and choice on overall dopamine/serotonin [line 196-205]:

This analysis indicated that overall levels of dopamine, but not serotonin, were modulated by social context. Specifically, while there were no choice-related effects on dopamine, there was a positive effect of condition, with higher dopamine in the human than the computer condition (pink above cyan in **Fig. 3a**, top; N-M1; choice, $\beta \pm SE = -0.03 \pm 0.10$, $t(454) = -0.29$, $p = .771$; condition, $\beta \pm SE = 0.85 \pm 0.28$, $t(454) = 3.04$, $p = .002$; choice x condition, $\beta \pm SE = -0.17 \pm 0.25$, $t(454) = -0.69$, $p = .493$). In other words, while dopamine may drive a general change in the willingness to accept an offer made by a human versus a computer (**Fig. 2a**), it does not drive individual choices per se. In contrast, there were no effects on serotonin (**Fig. 3a**, bottom; N-M1; choice, $\beta \pm SE = -0.06 \pm 0.11$, $t(454) = -0.57$, $p = .570$; condition, $\beta \pm SE = 0.00 \pm 0.29$, $t(454) = -0.01$, $p = .993$; choice x condition, $\beta \pm SE = -0.32 \pm 0.27$, $t(454) = -1.21$, $p = .227$).

(3.3) Serotonin and behavior in ultimatum game – this study uses a technique with high temporal resolution and high spatial focus – the authors examine serotonin levels in SNr. As neuromodulators have large spread effect in different locations in the brain, it is possible that previous works that used pharmacological interventions found effects that were driven by serotonin in other parts of the brain, as their method was not as spatially focused. Is it possible that the effects of serotonin and dopamine in SNr may differ from their effect in other brain regions during social interaction? Maybe it is worthwhile to

highlight the localized nature of the finding, and in general the idea that we should be more careful about stating roles of neuromodulators and neurotransmitters without the specific experimental context and specific brain region?

We thank the reviewer for prompting us to discuss whether the results are specific to the SNr or reflect dopamine/serotonin signals that are broadcasted widely within the brain – a great point which we should have considered.

Changes:

(3.3.A) We have added a paragraph to the Discussion which reviews the connections of the SNr and considers the regional specificity of the results [line 329-345]:

Our electrochemical data were collected from the SNr (**Fig. 1a**); we should therefore consider (1) its anatomical connections and (2) whether our results are specific to the SNr or reflect signals that are broadcasted widely within the brain. First, the SNr is one of the basal ganglia's main output nuclei: it receives excitatory glutamatergic inputs from the subthalamic nucleus⁵⁴, inhibitory GABAergic inputs from the striatum⁵⁴, dopaminergic inputs from substantia nigra pars compacta (SNc)⁵⁵, and serotonergic inputs from the raphe nucleus^{56,57}; and it sends GABAergic outputs to the thalamus⁵⁴ which control glutamatergic outputs from the thalamus – a main relay station for sensorimotor information – to cortical and sub-cortical regions^{54,58–61}. These distal projection targets include regions that support decision-making in non-social and social contexts, including orbitofrontal cortex, medial prefrontal cortex, anterior cingulate cortex, and the amygdala. Second, it is hard to say whether our results are specific to the SNr. Dopamine release in the SNr is mainly driven by somatodendritic release from the SNc⁵⁵, but this mechanism can be activated by action potentials in the SNc which drive synaptic release in other brain regions⁶². Similarly, while serotonin release in the SNr is mainly driven by direct synaptic release from the raphe nucleus^{56,57}, the upstream serotonergic neurons may project to other brain regions. Future research using human electrochemistry could offer insight about regional specificity by recording from multiple brain regions on the same task.

(3.4) The finding about dopamine and social context is very interesting, and current works tie dopamine levels to social intentions judgement, especially in schizophrenia and paranoia (see recent works by Vaughan Bell and Nichola Raihani, Joe Barnaby, Michael Moutoussis, as well as Jennifer Cook and others). I think that this finding could be better highlighted in the context of these works, besides the RPE finding.

We thank the reviewer for bringing this literature to our attention.

Changes:

(3.4.A) We now relate the effect of social context on overall dopamine levels to the literature on delusions in schizophrenia and biases in social reasoning in the Discussion [line 303-308]:

[...] One prediction of the hypothesis that dopamine helps set the stage for social interaction is that disturbances in dopamine signaling should increase the risk of social dysfunction. Indeed, schizophrenia, associated with a dysregulated dopamine system⁴⁵, can involve delusions centered around social themes (e.g., persecutory delusions)^{46,47}, sometimes ruining people's social lives. The hypothesis also fits with a growing literature linking dopamine to biases in social reasoning⁴⁸⁻⁵⁰, such as attribution of harmful intent. [...]

The new citations are included below for ease of reference:

45. Howes, O. D. & Kapur, S. The Dopamine Hypothesis of Schizophrenia: Version III--The Final Common Pathway. *Schizophrenia Bulletin* 35, 549–562 (2009).
46. Paget, A. & Ellett, L. Relationships among self, others, and persecutors in individuals with persecutory delusions: a repertory grid analysis. *Behav Ther* 45, 273–282 (2014).
47. Bell, V., Raihani, N. & Wilkinson, S. Derationalizing Delusions. *Clinical Psychological Science* (2020) doi:10.1177/2167702620951553.
48. Barnby, J. M., Bell, V., Deeley, Q. & Mehta, M. A. Dopamine manipulations modulate paranoid social inferences in healthy people. *Transl Psychiatry* 10, 1–13 (2020).
49. Barnby, J. M., Mehta, M. A. & Moutoussis, M. The computational relationship between reinforcement learning, social inference, and paranoia. *PLoS Comput Biol* 18, e1010326 (2022).

50. Schuster, B. A. et al. Dopaminergic modulation of dynamic emotion perception. *The Journal of Neuroscience* 42, 4394–4400 (2022).

Decision Letter, first revision:

20th December 2023

Dear Dr. Bang,

Thank you for your patience as we've prepared the guidelines for final submission of your Nature Human Behaviour manuscript, "Dopamine and serotonin in human substantia nigra track social context and value signals during economic exchange" (NATHUMBEHAV-23041343A). Please carefully follow the step-by-step instructions provided in the attached file, and add a response in each row of the table to indicate the changes that you have made. Please also check and comment on any additional marked-up edits we have proposed within the text. Ensuring that each point is addressed will help to ensure that your revised manuscript can be swiftly handed over to our production team.

We would hope to receive your revised paper, with all of the requested files and forms within two-three weeks. Please get in contact with us if you anticipate delays.

Nature Human Behaviour offers a Transparent Peer Review option for new original research manuscripts submitted after December 1st, 2019. As part of this initiative, we encourage our authors to support increased transparency into the peer review process by agreeing to have the reviewer comments, author rebuttal letters, and editorial decision letters published as a Supplementary item. When you submit your final files please clearly state in your cover letter whether or not you would like to participate in this initiative. Please note that failure to state your preference will result in delays in accepting your manuscript for publication.

In recognition of the time and expertise our reviewers provide to Nature Human Behaviour's editorial process, we would like to formally acknowledge their contribution to the external peer review of your manuscript entitled "Dopamine and serotonin in human substantia nigra track social context and value signals during economic exchange". For those reviewers who give their assent, we will be publishing their names alongside the published article.

Cover suggestions

We welcome submissions of artwork for consideration for our cover. For more information, please see our https://www.nature.com/documents/Nature_covers_author_guide.pdf target="new"> guide for cover artwork.

ORCID

Non-corresponding authors do not have to link their ORCIDs but are encouraged to do so. Please note that it will not be possible to add/modify ORCIDs at proof. Thus, please let your co-authors know that if they wish to have their ORCID added to the paper they must follow the procedure described in the following link prior to acceptance:

Nature Human Behaviour has now transitioned to a unified Rights Collection system which will allow our Author Services team to quickly and easily collect the rights and permissions required to publish your work. Approximately 10 days after your paper is formally accepted, you will receive an email in providing you with a link to complete the grant of rights. If your paper is eligible for Open Access, our Author Services team will also be in touch regarding any additional information that may be required to arrange payment for your article.

Please note that *Nature Human Behaviour* is a Transformative Journal (TJ). Authors may publish their research with us through the traditional subscription access route or make their paper immediately open access through payment of an article-processing charge (APC). Authors will not be required to make a final decision about access to their article until it has been accepted. Find out more about Transformative Journals

[REDACTED]

Best regards,
Abbey Ford
Editorial Assistant
Nature Human Behaviour

On behalf of

Giacomo Ariani
Editor
Nature Human Behaviour

Reviewer #1:

Remarks to the Author:

The authors have addressed my concerns in some detail and the responses are convincing.

I had one final thought about point 1.6 and the authors' response (itemised as 1.6.A in the additional_review_material). This was my biggest concern; i.e. the meaning of the trial-by-trial DA signals. Given the main disassociation of DA and 5-HT signalling and the additional analysis that

preserves the original findings but controls for absolute RPE, I'm not sure the authors need to say that '...the brain behavior dissociation.....does not mean that no learning took place'. The 'does not mean that no learning took place' seems to undersell the results unnecessarily.

In fact, the Xiang et al and Gu et al framing in terms of norm acquisition captures the kind of learning that I think likely happened in the experiment (and also partially answers my worry in 1.2). It's striking that the authors reference norms only twice across the first two paragraphs of the introduction but then not again until the final paragraph of the discussion. Personally, I think the manuscript would be improved by strengthened the norms in the discussion.

Reviewer #2:

Remarks to the Author:

The authors have addressed my previous concerns and provided a much more detailed description of the methodology used. I have no additional comments or suggestions.

Reviewer #3:

Remarks to the Author:

The authors answered all my questions to my satisfaction. I appreciate their revised discussion sections about localization of their observation in SNr, and the relation to evidence of relation between Dopamine and social context. Their clarification of statistical models is important for better understanding of the results.

I have no further comments.

Author Rebuttal, first revision:

Reviewer #1:

(1.1) The authors have addressed my concerns in some detail and the responses are convincing. I had one final thought about point 1.6 and the authors' response (itemised as 1.6.A in the additional_review_material). This was my biggest concern; i.e. the meaning of the trial-by-trial DA signals. Given the main disassociation of DA and 5-HT signalling and the additional analysis that preserves the original findings but controls

for absolute RPE, I'm not sure the authors need to say that '....the brain behavior dissociation.....does not mean that no learning took place'. The 'does not mean that no learning took place' seems to undersell the results unnecessarily. In fact, the Xiang et al and Gu et al framing in terms of norm acquisition captures the kind of learning that I think likely happened in the experiment (and also partially answers my worry in 1.2). It's striking that the authors reference norms only twice across the first two paragraphs of the introduction but then not again until the final paragraph of the discussion. Personally, I think the manuscript would be improved by strengthened the norms in the discussion.

We are pleased to hear that the reviewer feels we have addressed their concerns and thank them for their help with improving the manuscript. Following the reviewer's suggestion, we have rephrased several sentences in the Discussion to mention "norms" and have removed the sentences which "seems to undersell the results unnecessarily".

Changes:

(1.2.A) We now reference "norms" in the two opening paragraphs of the Discussion:

Previous work suggests that the dopamine and serotonin systems play central roles in human social interaction^{9,39}. However, because of methodological limitations, the contribution of these neuromodulators to social behavior has not yet been studied at fast timescales in humans. By applying a recently developed method for human electrochemistry during DBS surgery^{22–26}, we obtained sub-second estimates of dopamine and serotonin from the SNr while Parkinson's disease patients played the ultimatum game with both human and computer avatars. Despite receiving the same offers in both conditions, participants rejected more human than computer offers, **indicative of the human condition invoking social fairness norms**. The electrochemical data indicated that dopamine underpinned this behavioral response, with higher overall levels of dopamine, but not serotonin, in the human condition. Regardless of social context, and in support of a hypothesis that dopamine and serotonin carry distinct yet complementary value signals, changes in dopamine relative to a local baseline tracked trial-by-trial changes in offer value, whereas relative changes in serotonin tracked the current offer value. **Taken together, these results suggest that dopamine and serotonin support not only the computation of value statistics but also the norm-based use of these statistics during social interaction.**

Our behavioral data replicated the result that people reject more offers when they believe they are interacting with another person as opposed to a computer^{30–33}. This effect of social context, which is accompanied by increased affective arousal as measured by skin conductance³² and increased activity in emotion-related brain regions (e.g., amygdala, insula, and striatum)^{30,33}, has been attributed to human social interaction invoking a sense of fairness. While **rejecting "unfair" offers—enforcing fairness norms** can promote cooperation⁴⁰, research suggests that our sense of fairness is in fact self-oriented: we view unfair offers as displays of dominance and reject them to avoid the imposition of inferior status⁴¹ or gain social control⁴². Such a change in the frame of reference for social interaction may explain why overall

dopamine levels were higher for human than computer avatars. Indeed, pharmacological studies have found that elevated dopamine levels make people more averse to differences between their own and others' payoffs⁴³, less averse to inflicting pain on others in exchange for money¹⁴, and more selfish when selfish behaviors cannot be punished⁴⁴. One prediction of the hypothesis that dopamine helps set the stage for social interaction is that disturbances in dopamine signaling should increase the risk of social dysfunction. Indeed, schizophrenia, associated with a dysregulated dopamine system⁴⁵, can involve delusions centered around social themes (e.g., persecutory delusions)^{46,47}, sometimes ruining people's social lives. The hypothesis also fits with a growing literature linking dopamine to biases in social reasoning^{48–50}, such as attribution of harmful intent. While pharmacological studies have found a link between overall serotonin levels and the willingness to accept unfair offers^{10,11}, dietary acute tryptophan depletion does not influence neural discrimination between human and computer conditions in the ultimatum game as assessed by fMRI³⁰. In line with this result, we found no effect of social context on overall serotonin levels.

(1.2.B) We have removed the removed the sentences in the Discussion which "seems to undersell the results unnecessarily":

In addition to overall levels, we investigated how dopamine and serotonin changed relative to a local baseline, here the presentation of the current offer. Consistent with the RPE theory of dopamine^{15,16}, we found that relative changes in dopamine reflected the difference in value between the current and the previous offer: dopamine showed a relative decrease when value decreased (a negative RPE) and a relative increase when value increased (a positive RPE) regardless of the social context. This result from the human SNr fits with previous animal work which found that the activity of SNr neurons is indicative of modulation by RPEs⁵¹. In contrast, relative changes in serotonin reflected the value of the current offer, with a relative decrease for low values and a relative increase for high values regardless of the previous offer and the social context. Taken together, these response patterns indicate that dopamine and serotonin play complementary rather than opponent roles in value-based processes^{52,53} – with dopamine supporting a comparison of the present with the past and serotonin supporting an evaluation of the here and now – and that these roles generalize across contexts. We highlight that the brain-behavior dissociation — where dopamine, but not choice, was affected by task history — does not mean that no learning took place. Patients may have tracked offer values as indicated by the neural data but adopted fixed context-specific thresholds for offer acceptance.

Reviewer #2:

(2.1) The authors have addressed my previous concerns and provided a much more detailed description of the methodology used. I have no additional comments or suggestions.

We are pleased to hear that the reviewer feels we have addressed their concerns and thank

them for their help with improving the manuscript.

Reviewer #3:

(3.1) The authors answered all my questions to my satisfaction. I appreciate their revised discussion sections about localization of their observation in SNr, and the relation to evidence of relation between Dopamine and social context. Their clarification of statistical models is important for better understanding of the results. I have no further comments.

We are pleased to hear that the reviewer feels we have addressed their concerns and thank them for their help with improving the manuscript.

Final Decision Letter:

Dear Dr Bang,

We are pleased to inform you that your Article "Dopamine and serotonin in human substantia nigra track social context and value signals during economic exchange", has now been accepted for publication in Nature Human Behaviour.

Please note that *Nature Human Behaviour* is a Transformative Journal (TJ). Authors may publish their research with us through the traditional subscription access route or make their paper immediately open access through payment of an article-processing charge (APC). Authors will not be required to make a final decision about access to their article until it has been accepted. Find out more about Transformative Journals

Once your manuscript is typeset and you have completed the appropriate grant of rights, you will receive a link to your electronic proof via email with a request to make any corrections within 48 hours. If, when you receive your proof, you cannot meet this deadline, please inform us at

rjsproduction@springernature.com immediately. Once your paper has been scheduled for online publication, the Nature press office will be in touch to confirm the details.

With best regards,

Giacomo Ariani
Editor
Nature Human Behaviour